# Scaling Speculative Decoding with
# LOOKAHEAD REASONING

**Yichao Fu**[1]    **Rui Ge**[1*]   **Zelei Shao**[1*]   **Zhijie Deng**[2†]   **Hao Zhang**[1†]
[1]UCSD    [2]Shanghai Jiao Tong University

## Abstract

Reasoning models excel by generating long chain-of-thoughts, but decoding the resulting thousands of tokens is slow. Token-level speculative decoding (SD) helps, but its benefit is capped, because the chance that an entire $\gamma$-token guess is correct falls exponentially as $\gamma$ grows. This means allocating more compute for longer token drafts faces an algorithmic ceiling – making the speedup modest and hardware-agnostic. We raise this ceiling with LOOKAHEAD REASONING, which exploits a second, step-level layer of parallelism. Our key insight is that reasoning models generate step-by-step, and each step needs only to be semantically correct, not exact token matching. In LOOKAHEAD REASONING, a lightweight draft model proposes several future steps; the target model expands each proposal in one batched pass, and a verifier keeps semantically correct steps while letting the target regenerate any that fail. Token-level SD still operates within each reasoning step, so the two layers of parallelism multiply. We show LOOKAHEAD REASONING lifts the peak speedup of SD both theoretically and empirically. Across GSM8K, AIME, and other benchmarks, LOOKAHEAD REASONING improves the speedup of SD from 1.4x to 2.1x while preserving answer quality, and its speedup scales better with additional GPU throughput. Our code is available at `https://github.com/hao-ai-lab/LookaheadReasoning`

## 1    Introduction

Large reasoning models (LRMs) have recently pushed the state of the art in math problem solving and program synthesis by generating explicit, long Chains of Thoughts (CoT) [1]. In these models, an answer unfolds as a sequence of intermediate reasoning "steps", and each step arrives token by token via autoregressive decoding. If a solution needs $N$ steps and each step needs $T$ tokens, the model must generate $O(NT)$ tokens, often running into tens of thousands of tokens and minutes of wall-clock time. For instance, OpenAI's o1 model [2] may take more than 2 minutes to solve a single problem from the International Mathematical Olympiad (IMO) challenges.

Speculative decoding (SD) mitigates this token-level dependency by spending additional FLOPs to shorten the critical path of generation: a cheap draft model proposes $\gamma$ future tokens and the expensive target model then verifies them in parallel; if every guess matches, the decoding can fast forward $\gamma + 1$ positions at once. However, in the face of LRMs with long decode, two facts limit how far this idea can scale. First, the probability of an entire $\gamma$-token sequence is correct drops almost exponentially with $\gamma$ (§2), so the expected number of accepted tokens quickly saturates as $\gamma$ grows. Second, the verifier must still verify the target logits for all $\gamma$ positions, and that cost grows linearly. This results in a speedup curve that climbs with small $\gamma$, plateaus after a few dozen tokens, and can even decline once the verification cost dominates. For example, in a real profiling, we observe SD's speedup caps at 1.4x (Figure 2). As this ceiling is algorithmic rather than hardware-bound, this means

---

*Work done during an internship at UCSD.

†Corresponding authors.

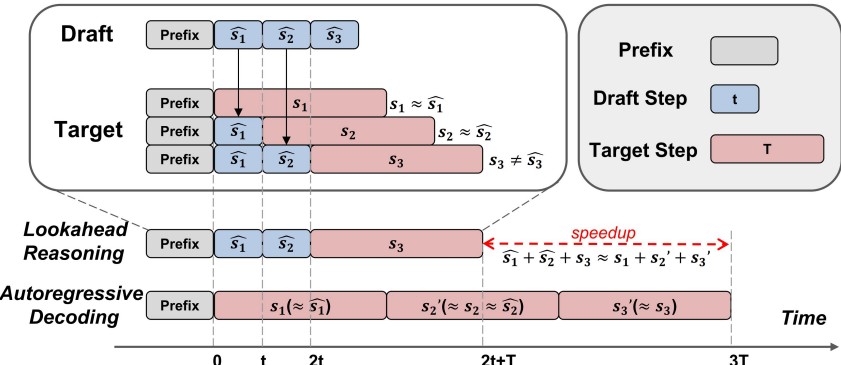

Figure 1: **One cycle of LOOKAHEAD REASONING.** The draft model proposes $\gamma = 3$ steps $\{\hat{s}_1, \hat{s}_2, \hat{s}_3\}$. The target model then generate $\{s_1, s_2, s_3\}$ based on prefixes and $\{\hat{s}_1, \hat{s}_2, \hat{s}_3\}$, respectively. Verifier checks if draft and target steps are semantically equivalent (e.g., $s_1 \approx \hat{s}_1$). If the first two steps are equivalent but the third is not, LOOKAHEAD REASONING outputs the verified draft steps $(\hat{s}_1, \hat{s}_2)$ followed by the target's correction $(s_3)$. This allows accepting multiple steps with only a lowered latency (e.g., $2t + T$) compared to the sequential target calls in autoregressive decoding (e.g., $3T$), where $t$ is draft step time and $T$ is target step time.

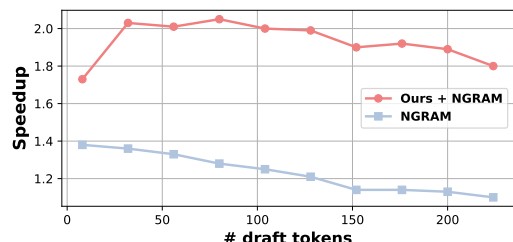

Figure 2: **Speedup vs Draft Tokens.** Speedup over autoregressive decoding, comparing LOOKA-HEAD REASONING combined with token-level SD (NGram-based) (red line) to SD alone (blue line). Our method is orthogonal to token-level SD and improves the maximum speedup from 1.4× to 2.1×.

that allocating more FLOPs in SD yields only diminishing returns, making SD's acceleration not scale with future accelerators. As LRMs produce ever-longer generations, the number of tokens SD can safely skip does not grow proportionally, so the end-to-end latency remains substantial.

This paper makes a key observation that reasoning is naturally hierarchical: a full chain-of-thought breaks into discrete steps, and each step unrolls tokens by token. To reach the correct answer, a reasoning step requires only semantic correctness but not exact token matches. To illustrate, we replaced over 50% of DeepSeek-R1 32B's reasoning steps with semantically equivalent ones from another smaller model. The impact on overall task accuracy was minimal, with deviations typically not exceeding 2% (§4.1). This looser requirement exposes a coarser unit for speculation: in addition to guessing the next few tokens, a model can guess and verify the next few reasoning steps. These step proposals and verification are independent, so they can be batched and executed in parallel, making full use of GPU's batching capacity. At the same time, token-level speculation can still operate within each step, achieving two complementary layers of parallelism rather than one.

This paper develops LOOKAHEAD REASONING based on this insight, with one operational cycle shown in Figure 1. First, a lightweight draft model autoregressively generates several sequential, future *reasoning steps* $\{\hat{s}_1, \hat{s}_2, \hat{s}_3\}$. Concurrently, the target LRM generates corresponding follow-up steps $\{s_1, s_2, s_3\}$, where each $s_i$ is generated based on a prefix formed by the initial context concatenated with the sequence of preceding draft steps $\hat{s}_1, \ldots, \hat{s}_{i-1}$. Notably, the generations of $\{s_1, s_2, s_3\}$ are issued as a batch running in parallel to exploit additional cross-request parallelism on GPUs. A lightweight verifier, implemented as a small LLM-as-a-Judge or an embedding model, then begins with the first speculative step to determine if the draft's original speculative step $\hat{s}_1$ semantically aligns with this oracle step $s_1$. If the step passes verification, we keep it and proceed

to the verification of $\hat{s}_2$ and $s_2$, which are already available due to batched execution. If it fails, we drop it and revert to the target model's standard generation. Concurrently, token-level speculation can operate independently when the target/draft model generates the content of each step.

LOOKAHEAD REASONING operates at step level, an axis orthogonal to token-level speculation. Because step-level speculation absorbs compute that would otherwise hit the speedup ceiling by token-level speculation, the method scales better with hardware. Additional FLOPs can draft more (or deeper) steps instead of lengthening token guesses, sidestepping diminishing returns faced by token-level-only speculative decoding. For example, on GSM8K, LOOKAHEAD REASONING lifts token-level SD's peak speedup from 1.4x to 2.1x (combined), as depicted in Figure 2. Even when available compute is limited, we prove that the peak speedup given limited compute shall be achieved only by combining both levels of speculation (§3.3).

A key design consideration is the choice of verifier. While an ideal semantic verifier ensures no accuracy loss, practical ones balance compute cost against judgment accuracy. For instance, a looser verification may boost draft acceptance (and speedup) but risks accuracy drop from erroneous steps. We finally opted for a 7B LLM-As-a-Judge, striking a balance between these competing factors (§4.3).

To sum up, our contributions can be listed as follows: (1) We develop LOOKAHEAD REASONING, a novel **step-level** speculation dimension to scale speculative decoding, orthogonal to existing **token-level** approaches. (2) We present theoretical analysis demonstrating significant speedups from our method, both as a standalone technique and when combined with token-level SD. (3) We conduct extensive experiments showing consistent performance improvements across diverse datasets.

## 2    Background

**Speculative Decoding.** LLMs autoregressively generate one token at a time, with each next token $x_{t+1}$ sampled from the distribution $P(x_{t+1} \mid x_{1:t})$. This sequential dependency poses a fundamental bottleneck to inference speed. Speculative decoding [3] mitigates this challenge using a "guess-and-verify" strategy with two models: a lightweight draft model $q$ and a strong target model $p$. Given a context $x_{1:t}$, $q$ autoregressively proposes a sequence of $\gamma$ candidates tokens, $\hat{x}_{t+1:t+\gamma}$, along with their draft probabilities $Q$. Subsequently, $p$ verifies these $\gamma$ tokens in a single parallel forward pass, yielding the target probabilities $P$. A rejection-sampling procedure then sequentially processes each proposed token $\hat{x}_{t+i}$. If a token is accepted, it is appended to the output; if rejected, the process halts, and a final token is sampled from $p$'s distribution based on the last accepted token. This allows for the acceptance of $n \le \gamma$ tokens in fewer steps than standard autoregression.

The theoretical speedup achieved by this speculative decoding approach, assuming negligible verification overhead beyond the target model's single pass, can be characterized by:

$$g(\gamma) = \frac{1 - \alpha^{\gamma+1}}{(1 - \alpha)(1 + c\,\gamma)},$$

where $\alpha$ represents the average acceptance rate of each token drafted by $q$ (assuming it's independent for each token in the sequence). Note that the probability of an entire sequence of $\gamma$ tokens being accepted typically decreases exponentially with $\gamma$, as errors accumulate. $c = T_q/T_p$ is the latency ratio of generating a single token by the draft model relative to the target model, where $T_q$ and $T_p$ are step time of $q$ and $p$, respectively. Furthermore, for all $c \ge 0$ and $\gamma \ge 0$, the speedup $g(\gamma)$ is upper-bounded by $1/(1 - \alpha)$, a limit approached only in the idealized case where $c = 0$. This inherent upper bound on $g(\gamma)$ signifies a critical limitation: beyond an optimal point, investing more computational resources by increasing the speculative length ($\gamma$) yields diminishing or even negative returns on speedup. Therefore, this algorithmic ceiling implies that the acceleration gains from token-level speculative decoding do not scale with improvements in hardware, such as more powerful GPUs. Consequently, for reasoning models with longer CoTs, the bounded acceleration offered by token-level speculative decoding alone highlights an urgent need for more potent acceleration strategies.

**LLM Reasoning.** Large reasoning models (LRMs) [2, 4] are increasingly pivotal for complex tasks such as math problem-solving and coding. These models often generate solutions by giving a "chain-of-thought" (CoT)—a sequence of intermediate reasoning steps, denoted as $s$, produced *step-by-step* to derive a final answer [1]. Each reasoning step ($s_i$) typically conditions on the previous one ($s_{i-1}$),

creating a sequential dependency analogous to token-level autoregression but at a higher conceptual granularity. We observe that this step-wise structure itself presents a significant opportunity for acceleration. Specifically, entire reasoning steps can be speculatively proposed by a draft model, denoted as $\hat{s}$. Our preliminary experiments (§4.1) show this potential. A small 1.5B draft model can generate speculative steps $\hat{s}$ that semantically align with over 50% of ground-truth steps $s$ from a much larger 32B target model. Besides, this is achieved while maintaining comparable task accuracy.

# 3 Method

In this section, we explain the details of LOOKAHEAD REASONING, then provide theoretical analysis that shows its performance benefits. Furthermore, since both step-level and token-level speculative generation rely on increasing concurrency, we show that in real-world settings—where the two methods compete for limited concurrency resources—peak performance gains can only be achieved when combining both speculative strategies together.

## 3.1 LOOKAHEAD REASONING: Step-Level Speculative Generation Solution

The core idea of LOOKAHEAD REASONING is to perform speculation and verification on entire steps rather than individual tokens. To put it clear, we first presented a synchronous version of this approach in Algorithm 1, and then conceptually illustrate an optimized asynchronous variant in Figure 1.

As detailed in Algorithm 1 (sync version), one cycle of LOOKAHEAD REASONING proceeds as follows:

1. **Draft Step Generation:** Given token prefix $x_{1:t}$, the draft model $q$ first autoregressive generates a sequence of $\gamma$ candidate steps, denoted as $\hat{s_0}, \ldots, \hat{s_{\gamma-1}}$. Each step $\hat{s_j}$ is generated conditioned on the prefix extended by all preceding draft steps: $x_{1:t} \oplus \bigoplus_{k=0}^{j-1} \hat{s_k}$. In practice, we simply use '\n\n' as the step break as we found it's a common flag for step split in various reasoning models [5, 4, 6].
2. **Parallel Target Step Generation:** Same as in the standard speculative decoding [3], once all $\gamma$ draft steps are available, target model $p$ generates following steps $s_0, \ldots, s_\gamma$ accordingly in parallel.
3. **Verification and Output Construction:** The algorithm then determines the longest prefix of accepted draft steps. Verification between each draft step $\hat{s_j}$ and its corresponding target step $s_j$ is performed by a verifier $V(s_j, \hat{s_j})$, which assesses whether if $\hat{s_j}$ is an acceptable substitute for $s_j$, i.e. whether they are semantic similar.

---

**Algorithm 1** LOOKAHEAD REASONING(Sync Version)

---

**Input:** Draft model $q$, Target model $p$, Prefix $x_{1:t}$, Max lookahead steps $\gamma$, Verifier $V(\cdot, \cdot) \to \{\text{True}, \text{False}\}$
  1: Initialize empty step sequences $\hat{s_0}, \ldots, \hat{s_{\gamma-1}}, s_0, \ldots, s_\gamma$
  2: $x_{\text{current}} \doteq x_{1:t}$
  3: **for** $j = 0$ to $\gamma - 1$ **do**                                      ▷ Generate $\gamma$ draft steps sequentially
  4:     $\hat{s_j} \doteq q.\text{GenerateStep}(x_{\text{current}}); \quad x_{\text{current}} \doteq x_{\text{current}} \oplus \hat{s_j}$
  5: **in parallel do** for $j = 0$ to $\gamma$:            ▷ Compute target steps $s_j$ in parallel based on draft prefixes
  6:     Let $x'_j \doteq x_{1:t} \oplus \bigoplus_{k=0}^{j-1} \hat{s_k}$ if $j \geq 1$ else $x_{1:t}$            ▷ Prefix before draft step $j$
  7:     $s_j \doteq p.\text{GenerateStep}(x'_j)$
  8: **end parallel**
  9: $j^* \doteq \min(\{j \in \{0..\gamma - 1\} \mid V(s_j, \hat{s_j}) == \text{False}\} \cup \{\gamma\})$            ▷ Find first unaccepted step using $V$
 10: OutputSequence $\doteq (\bigoplus_{k=0}^{j^*-1} \hat{s_k}) \oplus s_{j^*}$            ▷ Append verified drafts + decisive target
**Output:** OutputSequence

---

**Verifier Selection.** The choice of verifier ($V$) is a pivotal design consideration in LOOKAHEAD REASONING. While an ideal semantic verifier ensures no accuracy loss, practical implementations face a primary trade-off between judgment precision and computational overhead; Furthermore, the strictness of verification (e.g., a threshold) presents a secondary trade-off, potentially boosting draft acceptance and speedup at the risk of degrading task accuracy from erroneously accepted steps. We explore three common paradigms for semantic assessment—LLM-as-a-Judge [7] for nuanced evaluation, embedding-based verifier [8] for efficient similarity, and target model scoring [9]—each with distinct cost-precision profiles, empirically evaluating their impact in §4.3.

**Asynchronous Generation (Illustrated in Figure 1.** In Algorithm 1, the parallel verification steps launch only after all $\gamma$ draft steps $\hat{s}_j$ with $j \in \{0...\gamma - 1\}$ are produced. An optimized asynchronous implementation, conceptually depicted in Figure 1, can begin generating a target step $s_j$ as soon as its required prefix (containing $x_{1:t}, \hat{s}_0, \ldots, \hat{s}_{j-1}$) becomes available from the draft model. This async execution brings overlap for draft/target generation and verification phases, which can significantly reduce end-to-end latency compared with the synchronous version. Note that in this asynchronous mode implementation, both the draft and target models will "lookahead" $\gamma$ steps, ensuring maximal utilization of parallel processing. This is different from the sync version that draft model generate $\gamma$ drafts while target model generate $\gamma + 1$ steps in each cycle.

**Multi-Branch Drafting.** To further increase the number of the accepted reasoning steps, we explored tree-structure generation where the draft model proposes multiple candidate steps at each speculative position. Specifically, instead of generating a single candidate chain, the draft $q$ can propose a set of $W$ alternative steps for each position $j$ in the draft sequence. Once a step is generated, the draft then proposes $W$ child candidates in parallel for the subsequent position $j + 1$. This branching process continues up to a maximum $\gamma$ steps, leading to an exponential growth in the total number of candidate sequences explores, i.e., $W^\gamma$. The target model $p$, however, still generate one single candidate continuation step for each position $j$ (based on the draft prefix). The verifier $V$ would then check if *any* of the $W$ proposed draft branches for that position $j$ semantically aligns with the target model's step. If such a match is found, that branch is accepted and other branches are discarded. This multi-branch strategy aims to boost the likelihood of speculative success, albeit at the cost of increased computational effort in the drafting phase. We discussed its trade-off in §4.3.

### 3.2 Theoretical Speedup Analysis for LOOKAHEAD REASONING

We now analyze the potential performance gains of LOOKAHEAD REASONING. We make simplifying assumptions for clarity: negligible verification overhead, constant cost for generating steps, and a single draft branch at each stage.

**Notation:** Let $\gamma_1$ be the maximum number of draft steps generated sequentially, and $k_1 = \gamma_1$ be the number of target steps that generate in parallel. Let $T$ be the wall-time cost for the target model $p$ to generate one step, and $c_1 T$ be the cost for the draft model $q$ ($0 < c_1 < 1$). Let $\alpha_1 \in (0, 1)$ be the probability that a draft step is accepted.

**Step-Level Speedup for Sync Version of LOOKAHEAD REASONING:** The latency speedup for sync Lookahead Reasoning is

$$f_{sync}(k_1) = \frac{1 - \alpha_1^{k_1}}{(1 - \alpha_1)\,(1 - c_1 + c_1\,k_1)}.$$

**Step-Level Speedup for Async Version of LOOKAHEAD REASONING:** The speedup depends on whether the draft generation is limited by the maximum depth $k_1$ or by the relative cost $c_1$. Let $X_i$ be the number of consecutively accepted draft steps in stage $i$. The expected number of accepted steps before a rejection is $E[X_i] = \alpha_1/(1 - \alpha_1)$.

We define the asymptotic speedup $S$ as the ratio of steps generated by LOOKAHEAD REASONING compared to a target-only baseline over the same wall-time. Two cases arise:

1. $k_1 \geq \lceil 1/c_1 \rceil$: The draft model is relatively slow, and generating $k_1$ drafts takes longer than one target step. The depth limit $k_1$ is effectively inactive. The speedup converges to:

$$S_1 = \frac{1 + E[X_i]}{1 + c_1 E[X_i]} = \frac{1}{c_1 + (1 - c_1)(1 - \alpha_1)}$$

2. $k_1 < \lceil 1/c_1 \rceil$: The draft model is fast enough to generate $k_1$ steps within time $T$. The maximum depth $k_1$ limits the number of speculative steps per cycle. The speedup converges to:

$$S_2 = \frac{E[1 + X_i]}{E[\lceil (X_i + 1)/k_1 \rceil + c_1(X_i \bmod k_1)]} = \frac{1 - \alpha_1^{k_1}}{(1 - \alpha_1) + c_1 \big[\alpha_1 - \alpha_1^{k+1} - k_1(1 - \alpha_1)\alpha_1^{k_1}\big]}$$

(Detailed derivations are provided in Appendix B). Let $f_{async}(k_1)$ denote the step-level speedup, where $f_{async}(k_1) = S_1$ or $S_2$ depending on the case.

### 3.3 Optimal Speculation Strategies under Concurrency Constraints

Token-level speculative decoding [3] and step-level speculative decoding are orthogonal to each other. If $k_2$ is the number of *additional* tokens speculated by the draft model within each step generation (for both draft and target models, assuming they use internal speculation) and $c_2$ is the ratio of execution time of the draft model and target model in speculative decoding, its speedup is given by $g(k_2)$, based on the token acceptance rate $\alpha_2$:

$$g(k_2) = \frac{1 - \alpha_2^{k_2}}{(1 - \alpha_2)\,(1 - c + c\,k_2)}$$

Since these mechanisms operate at different granularities (inter-step vs. intra-step), their speedups multiply, yielding a combined speedup $h(k_1, k_2) = f(k_1) \times g(k_2)$. However, these two orthogonal parallelism dimensions compete for computational resources, making it crucial to determine the optimal resource allocation strategy to achieve maximum speedup. In this work, we focus on a fundamental question: is using both speculation methods superior to applying only one method?

**Optimality of Hybrid Approach under Budget Constraint:**

In real-world systems, memory and computational constraints necessitate capping the total degree of parallelism ($M$) available to the target model, i.e., $ParallelDim_g \times ParallelDim_f$ for step-level and token-level speculative decoding methods, respectively. This constraint transforms our earlier question into a resource allocation optimization problem: given a finite parallel budget ($M$), should we distribute resources across both parallelism dimensions or concentrate them on a single method? Consequently, our design goal becomes:

$$\max_{ParallelDim_g \times ParallelDim_f \leq M} h(k_1, k_2) = f(k_1) \times g(k_2). \tag{1}$$

$$\max_{\gamma_1 \times \gamma_2 \leq M} f(\gamma_1) \times g(\gamma_2). \tag{2}$$

Where $ParallelDim_g = k_2$, and

$$ParallelDim_f = \begin{cases} k_1, & sync \\ \min\{\lceil \frac{1}{c} \rceil, k_1\}, & async \end{cases}$$

It's easy to see that if we set $k_1 = 1$, then we are using purely token-level speculative decoding, whereas if we set $k_2 = 1$, then we are using purely lookahead reasoning.

**Theorem (Hybrid Method Optimality for Async Algorithm LOOKAHEAD REASONING):** Under the conditions of acceptance rates ($0.52 < \alpha_1, \alpha_2 < 0.8$), reasonably efficient draft models ($c_1 < \frac{1}{3}, c_2 < \frac{1}{5}$), and sufficient parallelism budget $M \geq 16$, the maximum speedup $h(k_1, k_2)$ is achieved if and only if a hybrid strategy is employed, meaning both $k_1 \geq 2$ and $k_2 \geq 2$.

These conditions are broadly representative of real-world scenarios: acceptance rates ($\alpha_1, \alpha_2$) in the 0.52-0.8 range are common in speculative decoding [10] and our experiments (§4.1); draft model efficiency ratios ($c_1$) below $\frac{1}{3}$ and ($c_2$) below $\frac{1}{5}$ are also common; and the parallelism budget ($M \geq 16$) reflects typical GPU capabilities. This analysis demonstrates that neither pure step-level nor pure token-level speculation is optimal under a fixed parallelism budget. The highest theoretical speedup is obtained by judiciously combining both strategies, leveraging parallelism across steps ($k_1$) and within steps ($k_2$). This motivates architectures and systems that can effectively manage and exploit both levels of speculative execution. It is empirically validated in §4.2. We provide a complete proof in Appendix B.1.2. Additionally, we analyze in detail the conditions under which single methods (either token-level or step-level) outperform hybrid approaches, and conversely, when combining both methods yields superior performance (Appendix B.1.2).

## 4 Experiment

*Models.* We evaluate two popular open-source reasoning model series: DeepSeek-R1-Distill [4] and Qwen3 [6]. For the DeepSeek-R1-Distill series, the 1.5B version serves as the draft model and the

32B version as the target model. Similarly, for the Qwen3 series, the 1.7B model is used as the draft model and the 32B model as the target. Unless otherwise specified, Qwen2.5-7B-Instruct [11] is employed as the judgement model. A deliberately designed judge prompt template allows our model to assess the semantic alignment between two sentences in just one prefill pass (Appendix A).

***Datasets.*** Our evaluation spans a suite of benchmarks, aligning with previous speculative decoding research [12, 13] and reasoning model evaluations [4, 6]. For code generation, we use HumanEval [14] and LiveCodeBench [15]. Math reasoning tasks are assessed using GSM8K [16], AIME'24 [17], and AMC12'23 [18]. For question answering, we include GPQA [19] and MT-Bench [7]. Specific to dataset sampling, we utilize 40 out of 50 problems from AMC12'23, selected by Qwen2.5 Math [20], and randomly sample 100 queries from the 1.3K GSM8K test set. For LiveCodeBench, We select 268 problems collected between August 2024 and Janaury 2025, following previous research [4].

***General Parameters.*** LLM generation settings are configured specifically for each model series. For the DeepSeek-R1-Distill series, we adhere to the official settings with a temperature of 0.6, top_p of 0.95, and a maximum generation length of 32K. For the Qwen3 series, the temperature is set to 0.6, top_p to 0.95, min_p to 0, top_k to 20, and the maximum generation length is 37K. These maximum generation lengths are chosen to ensure complete outputs. We use prompt-lookup decoding (n-gram) [21] as a representative speculative decoding (SD) method: the max lookup tokens is set to 1 for GSM8K and 2 for other datasets. The number of speculative tokens is set to 8 for SD and the number of speculative steps is set to 6 for LOOKAHEAD REASONING by default.

***Testbed.*** Experiments are conducted on a server equipped with eight NVIDIA H100 GPUs. Target models (32B) are deployed across two H100 GPUs using tensor parallelism. Draft models (1.5B/1.7B) and the default judge model (Qwen2.5-7B-Instruct) are each deployed on a single H100 GPU. Our algorithm is built upon the vLLM v0.8.3. Both the baseline and our proposed method are evaluated using vLLM [22] to simulate real-world LLM serving conditions.

***Evaluation Metrics.*** For code generation tasks (HumanEval, LiveCodeBench) and mathematical reasoning benchmarks (GSM8K, AIME'24, AMC12'23), we use accuracy (or pass@1). In question answering, accuracy is used for GPQA, while MT-Bench scores are obtained using Qwen2.5-72B-Instruct [11] as the judge. The accuracy on livecodebench is averaged over 8 samples while the accuracy on other datasets are averaged over 16 samples. We calculate the acceptance rate over the entire generation process and the accuracy of the final generated text. The evaluation procedure works as follows: at each generation step, we obtain outputs from both the draft and target models, then use a verifier to determine whether to accept or reject the draft output. The accepted result is added to the history trajectory, and this iterative process repeats until the end of generation is reached.

## 4.1 End-to-End Performance of LOOKAHEAD REASONING

We evaluated the end-to-end performance of LOOKAHEAD REASONING (LR) across diverse benchmarks using DeepSeek-R1-Distill and Qwen3 pairs. The detailed results are presented in Table 1. A key finding is LR's consistent ability to preseve task accuracy. Across a variety of benchmarks, LR's accuracy varies within a narrow range relative to the target model's autoregressive baseline, from approximately 1.0% above to 2.1% below baseline performance. This accuracy preservation contrasts with SpecReason, which exhibited more noticeable accuracy reductions on several tasks (e.g., dropping from $91.8\%$ to $85.9\%$ on GSM8K with Deepseek-R1, a $\sim 6\%$ decrease). This underscores LR's design principle of preserving output via robust semantic verification.

Furthermore, LR achieves strong accuracy while maintaining high step acceptance rates, often above 50% and reaching up to 63%. These substantial acceptance rates empirically support our initial insight that a smaller draft model can effectively predict semantically correct reasoning steps for a larger target model. LR also delivers significant efficiency gains. Its step-level parallelism is orthogonal to token-level speculative decoding, and their synergy produces substantial speedups. LR alone achieves speedups ranging from 1.04x to 1.71x across various benchmarks and model pairs. When combined with n-gram SD, the total speedup is further amplified, reaching up to 2.11x. This combined approach consistently outperforms n-gram SD alone, demonstrating the added value of step-level speculation. These results, consistent across both Deepseek-R1 and Qwen3 families, underscore the generalizable acceleration benefits of LR.

Table 1: LOOKAHEAD REASONING's Performance Across Datasets. Speedup is relative to the Autoregressive Decoding of the respective Target Model. We use $W = 1$ here.

| Method | Metric | Dataset | | | | | | |
|--------|--------|---------|---------|---------|-----------|---------|----------|--------------|
| | | AIME24 | AMC23 | GSM8K | HumanEval | GPQA | MT-Bench | LiveCodeBench |
| **Draft: Deepseek-R1-Distill 1.5B / Target: Deepseek-R1-Distill 32B** | | | | | | | | |
| Draft Model | Acc. (%) | $28.5 \pm 3.9$ | $71.6 \pm 4.1$ | $77.6 \pm 3.3$ | $67.2 \pm 2.4$ | $9.6 \pm 1.2$ | $6.23 \pm 1.9^*$ | $14.5 \pm 1.3$ |
| Target Mode | Acc. (%) | $70.8 \pm 5.2$ | $95.6 \pm 2.3$ | $91.8 \pm 1.9$ | $96.9 \pm 0.8$ | $63.3 \pm 2.2$ | $8.17 \pm 1.2^*$ | $48.9 \pm 1.3$ |
| SpecReason | Acc. (%) | $58.3 \pm 5.7$ | $90.6 \pm 2.6$ | $85.9 \pm 2.2$ | $94.5 \pm 1.5$ | $57.0 \pm 2.8$ | – | $40.6 \pm 1.5$ |
| | Apt. | 0.39 | 0.69 | 0.93 | 0.43 | 0.08 | – | 0.25 |
| LR(ours) | Acc. (%) | $69.2 \pm 8.1$ | $94.1 \pm 2.1$ | $92.8 \pm 1.8$ | $95.5 \pm 1.8$ | $61.2 \pm 2.8$ | $8.13 \pm 1.2^*$ | $49.5 \pm 2.3$ |
| | Apt. | 0.47 | 0.58 | 0.63 | 0.44 | 0.35 | 0.48 | 0.47 |
| | Speedup | 1.36× | 1.48× | 1.71× | 1.27× | 1.14× | 1.27× | 1.21× |
| SD | Speedup | 1.53× | 1.50× | 1.39× | 1.32× | 1.48× | 1.25× | 1.45× |
| SD+LR(ours) | Speedup | 1.82× | 2.00× | 2.11× | 1.54× | 1.63× | 1.51× | 1.58× |
| **Draft: Qwen3 1.7B / Target: Qwen3 32B** | | | | | | | | |
| Draft Model | Acc. (%) | $46.9 \pm 8.1$ | $84.2 \pm 4.7$ | $91.1 \pm 1.6$ | $85.4 \pm 1.6$ | $38.5 \pm 1.4$ | $7.96 \pm 1.5^*$ | $28.8 \pm 1.6$ |
| Target Model | Acc. (%) | $80.0 \pm 3.9$ | $97.5 \pm 2.0$ | $96.6 \pm 1.4$ | $97.6 \pm 0.8$ | $68.2 \pm 2.1$ | $8.53 \pm 1.1^*$ | $52.4 \pm 1.4$ |
| SpecReason | Acc. (%) | $68.3 \pm 5.3$ | $90.5 \pm 3.9$ | $94.5 \pm 1.4$ | $92.0 \pm 2.0$ | $66.3 \pm 2.0$ | – | $39.7 \pm 1.9$ |
| | Apt. | 0.75 | 0.92 | 0.95 | 0.91 | 0.46 | – | 0.65 |
| LR(ours) | Acc. (%) | $80.4 \pm 4.1$ | $96.4 \pm 2.0$ | $96.4 \pm 1.2$ | $97.1 \pm 0.8$ | $68.5 \pm 2.4$ | $8.46 \pm 1.15^*$ | $51.7 \pm 1.7$ |
| | Apt. | 0.43 | 0.53 | 0.50 | 0.39 | 0.30 | 0.38 | 0.40 |
| | Speedup | 1.12× | 1.22× | 1.32× | 1.13× | 1.04× | 1.10× | 1.08× |
| SD | Speedup | 1.40× | 1.38× | 1.32× | 1.32× | 1.40× | 1.41× | 1.25× |
| SD+LR(ours) | Speedup | 1.49× | 1.62× | 1.68× | 1.39× | 1.44× | 1.49× | 1.32× |

*Note*: LR (ours) refers to our proposed Lookahead Reasoning method. SD denotes token-level Speculative Decoding (N-gram based). Acc. stands for Accuracy (%), and Apt. for Acceptance Rate. For MT-Bench (marked with *), the reported metric is its standard score (0-9 scale) instead of accuracy. "–" indicates data not applicable. Speedup is relative to the autoregressive decoding of the respective target model.

## 4.2 Combining LOOKAHEAD REASONING with Speculative Decoding

To empirically validate the orthogonality of LOOKAHEAD REASONING (LR) with speculative decoding, we conducted experiments using prompt-lookup decoding (n-gram) on the AIME dataset.

Figure 3 shows the orthogonality of LR and Speculative Decoding (SD). Subplot (a) shows that while LR alone with varying draft step number reaches a speedup around 1.4x, adding SD boosts this to approximately 1.9x. Similarly, subplot (b) illustrates that SD alone with varying Speculative Token Numbers peaks around 1.55x speedup, but combining it with LR again achieves up to 1.9×. Collectively, these results highlight that while either method in isolation offers limited gains, their combination consistently yields the most significant performance improvements, aligning with our theoretical analysis in § 3.2.

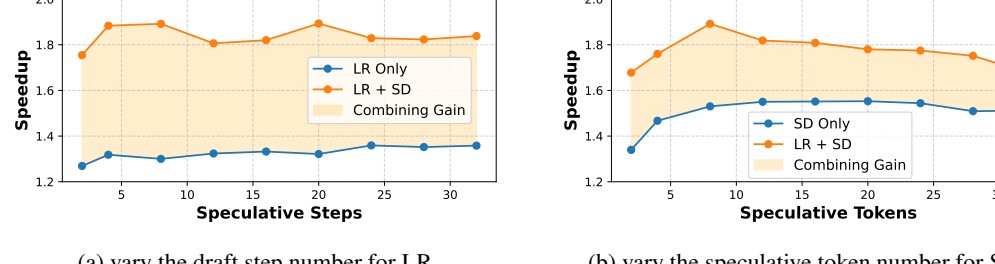

(a) vary the draft step number for LR      (b) vary the speculative token number for SD

Figure 3: **Orthogonality of Lookahead Reasoning and Speculative Decoding.** When used alone, the speedup from both LR and SD is limited by their draft length ($\gamma$). However, their combination consistently improves the max achievable speedup.

## 4.3 Ablation Study

**Effectiveness of the Verifier.** We conducted an ablation study to assess the impact of different verifier mechanisms on task accuracy, utilizing DeepSeek-R1-Distill 32B as the target model and a 1.5B parameter model as the draft model on GSM8K and AIME'24 datasets. We compare four verifiers: (1) **Random Acceptance** of drafts; (2) **LLM-as-a-Judge** (LLM-J) with Qwen2.5 7B/32B [11]; (3) **Embedding-based Verification** (Emb.) with all-mpnet-base-v2 model [8] at 0.85/0.95 similarity thresholds; and (4) **Reasoning Model Scoring**(Score) [9], using the target model to assign 0-9 scores with acceptance thresholds of 7/9. Results are in Table 2.

LLM-J verifiers (both 7B and 32B) showed robust accuracy preservation across both datasets, with minimal performance difference observed between the two verifier sizes. On GSM8K, their performance closely aligned with the original baseline, indicating no accuracy degradation. On AIME, LLM-J verifiers also maintained accuracy very close to the original, with observed deviations within approximately 1-2%. This contrasts sharply with Random Acceptance. Despite comparable or lower acceptance rates (e.g., 0.50 on GSM8K and 0.40 on AIME), Random Acceptance led to significant accuracy degradation on both GSM8K ($\sim 3.5\%$ lower) and AIME ($\sim 11\%$ lower). This underscores the necessity of a robust verification mechanism.

The Embedding-based verifier shows a trade-off: the stricter 0.95 threshold on GSM8K preserved accuracy ($92.3 \pm 1.4\%$) at a lower acceptance rate (0.37), while the 0.85 threshold, despite a higher acceptance rate (0.56), resulted in a $\sim 2\%$ accuracy drop ($89.8 \pm 2.5\%$). This pattern was mirrored on AIME. This indicates that while semantic equivalence is a promising criterion, its efficacy in preserving accuracy is highly dependent on the stringency (and precision) of the similarity judgement.

Reasoning Model Scoring, which assesses draft quality via target model scores rather than direct equivalence, consistently underperformed in accuracy preservation. For instance, even employing the stricter Threshold 9 resulted in notable accuracy reductions of approximately 5.9% on GSM8K and 12.5% on AIME. The still relatively high acceptance rate on GSM8K with this threshold (e.g., 0.93) suggests that the scoring mechanism may possess limited discriminative power on simpler datasets, even at stricter thresholds. This highlights a fundamental limitation: quality scores, even with high thresholds, do not reliably ensure alignment with the target model's output distribution, which is critical for LOOKAHEAD REASONING's correctness.

These results reveal that verifiers grounded in semantic equivalence with the target model's likely output are most effective for preserving accuracy within LOOKAHEAD REASONING. LLM-as-a-Judge excels in this, provding nuanced judgment, though with potential computational overhead. Embedding models provide a lightweight alternative (e.g., all-mpnet-base-v2 is only $\sim$100M parameters), where performance is tunable via the similarity threshold, offering a cost-effective solution.

Table 2: Performance comparison with different verifiers on GSM8K and AIME datasets. Apt.: accept rate; Acc.: accuracy (%).

| Dataset | Metric | Orig. | Rand. | LLM-J (Qwen) | | Emb. (Th.) | | Score (Th.) | |
|---|---|---|---|---|---|---|---|---|---|
| | | | | 7B | 32B | 0.85 | 0.95 | 7 | 9 |
| GSM8K | Apt. | − | 0.50 | 0.63 | 0.58 | 0.56 | 0.37 | 0.97 | 0.93 |
| | Acc. | $91.8 \pm 1.9$ | $88.3 \pm 3.7$ | $92.8 \pm 1.8$ | $92.3 \pm 1.2$ | $89.8 \pm 2.5$ | $92.3 \pm 1.4$ | $82.1 \pm 2.4$ | $85.9 \pm 2.2$ |
| AIME | Apt. | − | 0.40 | 0.47 | 0.46 | 0.45 | 0.38 | 0.81 | 0.39 |
| | Acc. | $70.8 \pm 5.2$ | $59.6 \pm 5.4$ | $69.2 \pm 8.1$ | $69.0 \pm 4.7$ | $64.0 \pm 6.5$ | $66.7 \pm 6.3$ | $37.9 \pm 7.5$ | $58.3 \pm 5.7$ |

**Effect of Tree Width on Performance** We investigate the impact of speculation tree width on LOOKAHEAD REASONING's accuracy, accept rate, and speedup, keeping depth $\gamma = 2$. In LOOKAHEAD REASONING, candidate sequences grow as $W^\gamma$. Wider trees ($W > 1$) can boost accept rate but escalate FLOPs and, with imperfect verifiers, risk accuracy degradation due to erroneous acceptances. We hypothesize a stronger verifier mitigates this. Experiments on GSM8K and AIME24 used Qwen2.5-7B-Instruct and Qwen2.5-32B-Instruct as judges. Results are demonstrated in Table 3.

Increasing $W$ consistently raised accept rate across datasets and judges (e.g., GSM8K with Qwen2.5-7B: accept rate 0.63 for $W = 1$ to 0.83 for $W = 8$). However, this rarely translated to better speedup beyond $W = 2$. Further widening often diminished speedup (e.g., AIME24 with Qwen2.5-7B, $W = 8$ yielded no speedup), likely due to the exponential overhead outweighing accept rate gains. Accuracy trends highlight verifier importance. With the Qwen2.5-7B judge, increasing

$W$ led to a noticeable accuracy drop, especially on AIME24 (from 69.2% at $W = 1$ to 64.6% at $W = 8$), supporting our hypothesis. The stronger Qwen2.5-32B judge demonstrated greater resilience: accuracy remained more stable on GSM8K, and the degradation on AIME24 was less pronounced (69.0% at $W = 1$ to 67.3% at $W = 8$). This indicates a stronger verifier is crucial for wider trees to manage the increased risk of incorrect speculation.

Table 3: Impact of Tree Width ($W$) on Performance Metrics (Depth $\gamma = 2$)

| Dataset | Judge | W=1 | | | W=2 | | | W=4 | | | W=8 | | |
|---|---|---|---|---|---|---|---|---|---|---|---|---|---|
| | | Acc.(%) | Apt. | Spd. | Acc.(%) | Apt. | Spd. | Acc.(%) | Apt. | Spd. | Acc.(%) | Apt. | Spd. |
| GSM8K | Qwen7B | $92.8 \pm 1.8$ | 0.63 | 1.48× | $91.2 \pm 1.8$ | 0.73 | 1.49× | $91.1 \pm 1.7$ | 0.77 | 1.47× | $91.5 \pm 1.8$ | 0.83 | 1.25× |
| | Qwen32B | $92.3 \pm 1.2$ | 0.58 | 1.40× | $93.2 \pm 2.0$ | 0.66 | 1.42× | $92.8 \pm 1.8$ | 0.73 | 1.39× | $92.5 \pm 1.5$ | 0.77 | 1.19× |
| AIME24 | Qwen7B | $69.2 \pm 8.1$ | 0.47 | 1.27× | $67.3 \pm 4.1$ | 0.58 | 1.32× | $65.4 \pm 6.5$ | 0.67 | 1.26× | $64.6 \pm 5.9$ | 0.74 | 1.00× |
| | Qwen32B | $69.0 \pm 4.7$ | 0.46 | 1.23× | $69.0 \pm 6.7$ | 0.54 | 1.23× | $68.1 \pm 6.1$ | 0.59 | 1.17× | $67.3 \pm 7.1$ | 0.68 | 0.98× |

*Note: Acc.: Accuracy (%); Apt.: Accept Rate; Spd.: Speedup (relative to target model autoregressive decoding). Depth $\gamma = 2$. Qwen2.5-7B/32B-Instruct are judge models.*

## 5  Related Work

**Speculative Decoding.** There are many different types of speculative decoding approaches. Draft-head methods like Medusa [23], Hydra [24], and EAGLE [25, 10, 13] integrate auxiliary heads into the target model to propose sequences. In contrast, Jacobi-based approaches such as Lookahead Decoding [12] and CLLM [26] enable parallel n-gram generation without draft models. System-level efforts [27, 28] further optimize SD's runtime efficiency in serving systems. LOOKAHEAD REASONING introduces a complementary form of step-level speculation tailored for reasoning models, enabling a new dimension of parallelism orthogonal to token-level methods.

**LLM Reasoning.** Recent trends shift from scaling model size [29, 30] to scaling inference-time compute [31, 32, 33], enabling large reasoning models (LRMs) like OpenAI o3/o4 [5], Kimi-K1.5 [34], and DeepSeek-R1 [4] to generate longer CoT to solve more complex problems in many steps. Recent work has begun to leverage this inherent step-by-step structure to accelerate LRMs. For instance, Speculative Thinking [35] uses LRMs to guide a smaller draft model, while SpecReason [9] accelerates reasoning by employing the large model to score the output of a small model, thereby deciding whether to accept its generated step. However, unlike LOOKAHEAD REASONING, these methods do not pursue a step-level equivalent with the original target model to accelerate reasoning.

## 6  Limitation and Conclusion

This paper introduces LOOKAHEAD REASONING, a novel method for accelerating large reasoning models during long CoT reasoning. LOOKAHEAD REASONING adds a new step-level dimension of parallelism, complementing traditional token-level speculative decoding. Our approach uses a draft model to propose future reasoning steps, which are then semantically verified by the target model. Evaluation on various datasets using two open-source reasoning models show that it can achieve up to 2.1X speedup combined with speculative decoding. This highlights LOOKAHEAD REASONING's effectiveness in making large reasoning models faster. This work has limitations that suggest future improvements. First, using '\n\n' to split reasoning steps is simple but may miss optimal breaks; smarter segmentation is needed. Second, current verifiers still trade speed for accuracy; designing lightweight but robust semantic verifiers remains an open challenge. Third, LOOKAHEAD REASONING introduces non-trivial implementation complexity, as integrating lookahead execution into production serving engines (e.g., SGLang or vLLM) requires additional scheduling logic to manage the step hierarchy. Some hyperparameters, such as $W$ and $\gamma$, also require tuning to balance aggressiveness and stability, which may introduce additional engineering overhead in deployment. While LOOKAHEAD REASONING improves per-request latency by batching candidate steps, it increases instantaneous GPU utilization and KV cache usage. This trades computation for speed and may reduce overall throughput under fully saturated GPU conditions. However, like other speculative decoding methods, it is particularly beneficial in latency-sensitive serving scenarios or when spare compute capacity is available.

## Acknowledge

We sincerely thank the anonymous reviewers for their insightful and constructive feedback. We gratefully acknowledge MBZUAI and NVIDIA for providing the computational resources that made this work possible, and we thank Yiming Zhao for his contributions in open-sourcing the project. Z. Deng is partially supported by NSF of China (Nos. 92470118, 62306176) and Natural Science Foundation of Shanghai (No. 23ZR1428700).

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

## A Judgement Prompt Template

---
**Semantic Equivalence Analysis Prompt**

<|im_start|>system
You are Qwen, created by Alibaba Cloud. You are a helpful assistant.<|im_end|>
<|im_start|>user
Evaluate whether the following two reasoning steps (s1 and s2) convey exactly the same meaning. Focus on semantic similarity rather than exact wording.
Compare the main ideas, key points, overall message, logical structure, and numerical calculations/results of both reasoning steps.
If the reasoning steps convey essentially the same meaning and generate same calculation results, respond with [aligned]. If the reasoning steps express different meanings, respond with [unaligned]. If it is too hard to determine, respond with [unaligned]
Please directly provide the final result in [aligned] or [unaligned].
Reasoning step 1 (s1):
<start_s1>
{}
<end_s1>
Reasoning step 2 (s2):
<start_s2>
{}
<end_s2><|im_end|>
<|im_start|>assistant
[
---

This prompt template is specifically designed for Qwen2.5 Instruct models. It guides the model to directly output either "aligned" or "unaligned" as its judgment. Consequently, a user can determine semantic equivalence between two sentences (s1 and s2) by simply checking if the model's initial output string begins with "ali" (the first three letters of "aligned"). Thus, only one forward pass is needed to get the result and judgement latency can be largely saved.

# B    Detailed Speedup Analysis

## B.1    Performace Gains Analysis

### B.1.1    Speedup Analysis of Async LOOKAHEAD REASONING

For the analysis that follows, we assume all sequences are of equal length and that the draft tree contains exactly one branch at each layer. Moreover, we treat the verifier's overhead as negligible.

**Notation.**

- $\gamma \in \mathbb{N}$: maximum number of generations the large model performs in parallel.
- $T$: cost (wall-time) of one sentence generated by target-model.
- $c_1$: cost of one draft-model run, measured in units of $T$ (so drafting costs $c_1 T$).
- $\alpha_1 \in (0, 1)$:accept rate of the drafts
- $X_i$: number of *consecutive* accepted drafts in stage $i$ before the first rejection.

We view a full generation as a sequence of DRAFT STAGE, each of which proceeds as follows:

1. If the number of generations the large model performs in parallel is less than $\gamma$. The draft model sequentially generate drafts.
2. Each time when we start to generate a draft step, we immediately ask the target model to generate a target step.
3. After the target model finished generation, immediately ask the verifier to verify whether should we accept the draft. If the draft was reject, fall back to the target model's original sentence and proceed to the next DRAFT STAGE.

Since each draft is accepted independently,

$$P(X_i = k) \ = \ \alpha^k (1 - \alpha), \qquad E[X_i] \ = \ \frac{\alpha}{1 - \alpha}.$$

**Theorem 1.** *The latency speedup for sync* LOOKAHEAD REASONING *is*

$$f_{sync}(\gamma) = \frac{1 - \alpha^{\gamma}}{(1 - \alpha)(1 - c + c\gamma)}.$$

*Proof.* The proof follows the same reasoning as in [3]. The only difference is that our $\gamma$ represents the maximum number of tokens the large model generates in parallel, whereas in their notation, it corresponds to $\gamma + 1$    □

**Theorem 2.** *Let*

$$S \ = \ \frac{\text{total sentences generated by our algorithm in } n \text{ DRAFT STAGE}}{\text{total sentences generated by target-only model in } n \text{ DRAFT STAGE}},$$

*We can see this as the latency speedup using async* LOOKAHEAD REASONING *algorithm. Then:*

1. *If $\gamma \geq \lceil \frac{1}{c_1} \rceil$, the draft tree never saturates. The parallel dimension of the target model is $\lceil \frac{1}{c_1} \rceil$, and as $n \to \infty$, the asymptotic speedup is*

$$S_1 \ = \ \frac{1}{c_1 + (1 - c_1)(1 - \alpha_1)}.$$

2. *If $\gamma < \lceil \frac{1}{c_1} \rceil$, the draft tree is depth-limited. The parallel dimension of the target model is $\gamma$, and as $n \to \infty$, the asymptotic speedup is*

$$S_2 \ = \ \frac{1 - \alpha_1^{\gamma}}{(1 - \alpha_1) + c_1 \big[ \alpha_1 - \alpha_1^{\gamma+1} - \gamma(1 - \alpha_1)\alpha_1^{\gamma} \big]} \ .$$

*Proof.* Over $n$ stages, we compare the total number of sentences generated by our algorithm to that produced by the baseline (target-only) approach.

**Case 1** When $\gamma \geq \left\lceil \frac{1}{c_1} \right\rceil$, note that since each draft costs $c_1 T$, the draft model can generate at most $\left\lceil \frac{1}{c_1} \right\rceil$ sentences during the time $T$ required for the target model to produce one sentence. Therefore, the draft tree never saturates, and the parallel dimension of the target model is effectively $\left\lceil \frac{1}{c_1} \right\rceil$.

Moreover, in DRAFT STAGE $i$ our algorithm spends

$$T + c_1 T X_i$$

total time. Over that same interval, the baseline target-only model would have produced

$$\frac{T + c_1 T X_i}{T} = 1 + c_1 X_i$$

sentences, while our algorithm emits $1 + X_i$ sentences. Thus over $n$ stages, according to the Law of Large Numbers,

$$S_1(n) = \frac{\sum_i (1 + X_i)}{\sum_i (1 + c_1 X_i)} = \frac{n + \sum_i X_i}{n + c_1 \sum_i X_i} \xrightarrow[n \to \infty]{} \frac{1 + E[X_i]}{1 + c_1 E[X_i]} = \frac{1}{c_1 + (1 - c_1)(1 - \alpha_1)}.$$

**Case 2** When $\gamma < \lceil \frac{1}{c_1} \rceil$, the draft-tree saturates at depth $\gamma$. So the parallel dimension of the target model would be $\gamma$ To emit a total of $X_i + 1$ sentences in stage $i$, we therefore proceed in

$$\left\lceil \frac{X_i + 1}{\gamma} \right\rceil$$

full intervals of length $T$, plus a final partial batch of size $X_i \bmod \gamma$. Hence the total wall-time for stage $i$ is

$$\left\lceil \frac{X_i + 1}{\gamma} \right\rceil T + c_1 T \left( X_i \bmod \gamma \right).$$

Over that same interval, the baseline target-only model would have generated

$$\left\lceil \frac{X_i + 1}{\gamma} \right\rceil + c_1 \left( X_i \bmod \gamma \right)$$

sentences, whereas our algorithm emits $1 + X_i$. Therefore, as $n \to \infty$

$$S_2(n) = \frac{n + \sum_{i=1}^n X_i}{\sum_{i=1}^n \left\lceil \frac{X_i + 1}{\gamma} \right\rceil + c_1 \sum_{i=1}^n (X_i \bmod \gamma)} \longrightarrow \frac{1 - \alpha_1^\gamma}{(1 - \alpha_1) + c_1 \left[ \alpha_1 - \alpha_1^{\gamma+1} - \gamma(1 - \alpha_1)\alpha_1^\gamma \right]}.$$

We put the calculation details in the appendix. $\qquad \square$

### B.1.2 Optimal Speculation Strategies under Concurrency Constraints

In this section we show that under a fixed parallelism budget, the optimal inference speedup is always achieved by jointly applying step-level and token-level parallelism, rather than by using either in isolation. Concretely, let $\gamma_2$ denote the degree of parallelism used by speculative decoding and $c_2$ be the ratio of the per-step execution time of the draft model to that of the target model. Then token-level speculative decoding alone yields[3]

$$g(\gamma_2) = \frac{1 - \alpha_2^{\gamma_2}}{(1 - \alpha_2)(1 - c_2 + c_2 \gamma_2)}. \tag{3}$$

(Note: here the formula is a little different than the one in [3] due to different definition.) Next, a pure step-level asynchronous parallelism scheme of depth $\gamma_1$ achieves speedup

$$f_{async}(\gamma_1) = \begin{cases} S_2 = \dfrac{1 - \alpha_1^{\gamma_1}}{1 - \alpha_1 + c_1 \alpha_1 (1 - \alpha_1^{\gamma_1}) - c_1 \gamma_1 \alpha_1^{\gamma_1} (1 - \alpha_1)}, & \gamma_1 < \lceil \frac{1}{c_1} \rceil \\ S_1 = \dfrac{1}{c_1 + (1 - c_1)(1 - \alpha_1)}, & \text{otherwise} \end{cases}$$

and a pure step-level synchronous parallelism scheme of depth $\gamma_1$ achieves speedup

$$f_{sync}(\gamma_1) = \frac{1 - \alpha_1^{\gamma_1}}{(1 - \alpha_1)(1 - c_1 + c_1 \gamma_1)}. \tag{4}$$

When both schemes are combined, the resulting speedup factorizes:

$$f(\gamma_1) \times g(\gamma_2). \tag{5}$$

Since in real-world systems, due to memory or compute constraints, we often need to cap the total degree of parallelism i.e. $\gamma_1\gamma_2$ in a hybrid speculative setting to $M$. In this case, we may ask that given a finite parallel budget $M$, is it better that we combine these two parallel dimensions or we only use one of them? Thus our design goal becomes

$$\max_{ParallelDim_g \times ParallelDim_f \leq M} h(\gamma_1, \gamma_2) = f(\gamma_1) \times g(\gamma_2). \tag{6}$$

Where $ParallelDim_g = \gamma_2$, and

$$ParallelDim_f = \begin{cases} \gamma_1, & sync \\ \min\{\lceil \frac{1}{c} \rceil, \gamma_1\}, & async \end{cases}$$

It's easy to see that if we set $\gamma_1 = 1$, then we are using purely token-level speculative decoding, whereas if we set $\gamma_2 = 1$, then we are using purely LOOKAHEAD REASONING.

**Theorem 3.** *Under synchronous* LOOKAHEAD REASONING *with concurrency budget $M \geq 4$, M is an even number, and the mild parameter constraint*

$$\min\left\{\frac{1 + \alpha_1}{1 + c_1}, \frac{1 + \alpha_2}{1 + c_2}\right\} > 1.157, \tag{7}$$

*Then at least one of the following must hold:*

$$\frac{1 + \alpha_1}{1 + c_1} \geq \frac{(1 + \alpha_2^{M/2})(1 - c_2 + \frac{c_2 M}{2})}{1 - c_2 + c_2 M}, \tag{8}$$

$$\frac{1 + \alpha_2}{1 + c_2} \geq \frac{(1 + \alpha_1^{M/2})(1 - c_1 + \frac{c_1 M}{2})}{1 - c_1 + c_1 M}. \tag{9}$$

*Furthermore, if both* (8) *and* (9) *hold simultaneously, then combining both speculative techniques strictly outperforms using either one alone. Conversely:*

- *If* (8) *fails, the optimal strategy is to use only token-level speculation.*

- *If* (9) *fails, the optimal strategy is to use only step-level speculation.*

*Proof.* **Step 1: At least one of** (8) **and** (9) **must hold** Define, for $i = 1, 2$,

$$D_i(M) = (1 + \alpha_i^{M/2}) \frac{1 - c_i + \frac{c_i M}{2}}{1 - c_i + c_i M}.$$

Observe that both factors

$$1 + \alpha_i^{M/2} \quad \text{and} \quad \frac{1 - c_i + \frac{c_i M}{2}}{1 - c_i + c_i M}$$

are strictly decreasing in $M$. Hence each $D_i(M)$ decreases as $M$ grows. In particular,

$$D_i(2) = \frac{1 + \alpha_i}{1 + c_i}.$$

Since either $D_1(2) \geq D_2(2)$ or $D_1(2) < D_2(2)$, it follows that either

$$D_1(2) > D_2(M) \implies \frac{1 + \alpha_1}{1 + c_1} > D_2(M),$$

or

$$D_2(2) > D_1(M) \implies \frac{1 + \alpha_2}{1 + c_2} > D_1(M).$$

In other words, at least one of (8) or (9) must hold.

**Step 2:** **Token-level-only speculation is suboptimal if condition** (8) **holds; otherwise, it is the optimal strategy.** Lemma 3 shows that both $f_{sync}(\gamma_1)$ and $g(\gamma_2)$ are unimodal, reaching their maxima at $\gamma_1^* \geq 2$ and $\gamma_2^* \geq 2$. Below, we analyze whether token-level-only speculation (i.e. $\gamma_1 = 1$) yields the best speedup for different ranges of the concurrency budget $M$.

**Case 1: If $M < \gamma_2^*$** In this case, when (8) holds, we have $h(1, M) \leq h(2, \frac{M}{2})$(transform through formulas), then according to the monotonicity of $h$, we have

$$h(1, \gamma_2) \leq h(1, M) \leq h(2, \frac{M}{2}), \forall 1 \leq \gamma_2 \leq M$$

Therefore, the overall maximum is attained only by jointly employing both levels. However, when (8) fails, since $D_i(x)$ is strictly decreasing, we know that

$$\frac{1 + \alpha_1}{1 + c_1} < \frac{(1 + \alpha_2^{x/2})(1 - c_2 + \frac{c_2 x}{2})}{1 - c_2 + c_2 x} \forall x \leq M$$

In this case, according to Lemma 5,

$$h(\gamma_1, \gamma_2) \leq h(1, \gamma_1 \gamma_2) \leq h(1, M), \forall \gamma_1, \gamma_2 \geq 1, \gamma_1 \gamma_2 \leq M$$

So the overall maximum is attained when we use token-level-only technique.

**Case 2: If $\gamma_2^* \leq M < 2\gamma_2^*$** In this case, we first prove that

$$h(1, \gamma_2^*) < h(2, \frac{\gamma_2^*}{2}), \forall 1 \leq \gamma_2 \leq M$$

Then we prove that 8 is always hold. From above we can see

$$h(1, \gamma_2) \leq h(1, \gamma_2^*) < h(2, \frac{\gamma_2^*}{2}), \forall 1 \leq \gamma_2 \leq M$$

And the theorem follows.

1.From Lemma3, we can know that

$$a_2(\gamma_2^*) = \frac{\alpha_2^{\gamma_2^*}(-\ln \alpha_2^{\gamma_2^*})}{1 - \alpha_2^{\gamma^*}} = \frac{c_2 \gamma_2^*}{1 - c_2 + c_2 \gamma_2^*}$$

Therefore

$$h(2, \frac{\gamma_2^*}{2})/h(1, \gamma_2^*) = \frac{1 + \alpha_1}{1 + c_1} \frac{1 - c_2 + c_2 \gamma_2^*}{(1 + \alpha_2^{\gamma_2^*/2})(1 - c_2 + c_2 \frac{\gamma_2^*}{2})}$$

$$= \frac{1 + \alpha_1}{1 + c_1} \frac{1}{1 + \alpha_2^{\gamma_2^*/2}} \frac{1}{1 - \frac{1}{2} a_2(\gamma_2^*)}$$

Let $y = 1 - \alpha_2^{\gamma_2^*/2} \in (0, 1)$,

$$(1 + \alpha_2^{\gamma_2^*/2})(1 - \frac{1}{2}a_2(\gamma_2^*)) = 2 - y + (y - 2 + \frac{1}{y}) \ln (1 - y)$$

Then from Lemma 6, we can see that its value was less than 1.157. Then given 7 we can have

$$h(2, \frac{\gamma_2^*}{2})/h(1, \gamma_2^*) > 1$$

2.From previous step, we know that $D_2(M)$ is strictly decreasing, so

$$D_2(M) \leq D_2(\gamma_2^*) < 1.157 < \frac{1 + \alpha_1}{1 + c_1}$$

So 8 is always hold.

**Case 3: If $M \geq 2\gamma_2^*$** In this case, same as in Case 2, 8 is always hold. Besides, we have

$$h(1, \gamma_2) \leq h(1, \gamma_2^*) < h(2, \gamma_2^*), \forall 1 \leq \gamma_2 \leq M$$

So it's obvious token-level-only speculation is suboptimal.

**Step 3: If** (9) **holds, step-level-only speculation is suboptimal. Otherwise, it is the optimal strategy.** Same as the previous step.

$\square$

**Theorem 4.** *Under asynchronous* LOOKAHEAD REASONING *with* $0.52 < \alpha_1, \alpha_2 < 0.8$ *and* $c_1 < \frac{1}{3}, c_2 < \frac{1}{5}$ *and the total parallelism budget* $M \geq 16$ *and M is an even number. Then under constraint* $\min\{\lceil \frac{1}{c_1} \rceil, \gamma_1\} \times \gamma_2 \leq M$*the overall speedup* $h(\gamma_1, \gamma_2)$ *is maximized if and only if both step-level and token-level parallelism are employed (i.e.* $\gamma_1 \geq 2$ *and* $\gamma_2 \geq 2$*).*

*Proof.* **Step 1: Token-level-only is not optimal.** Lemma 3 shows that $g(\gamma_2)$ is unimodal and reaching their maxima at $\gamma_2^* \geq 2$.

**1.**$M \geq 2\gamma_2^*$ This case, we have

$$h(1, \gamma_2) \leq h(1, \gamma_2^*) \leq h(2, \gamma_2^*), \forall 1 \leq \gamma_2 \leq M$$

Therefore, it's obvious that token-level-only is not optimal.

**2.**$\gamma_2^* \leq M < 2\gamma_2^*$ In this case, we only need to prove

$$h(1, \gamma_2^*) \leq h(2, \frac{\gamma_2^*}{2})$$

Same as the proof in 3, we only need to prove that

$$\frac{1 + \alpha_1}{1 + c_1} \geq 1.157$$

is hold, and it's easy to prove.

**3.**$M < \gamma_2^*$ This case, according to Lemma 1, we can see that

$$h(1, \gamma_2) < h(2, \frac{\gamma_2}{2})$$

is always hold, so we can conclude that tokne-level-only is not optimal.

**Step 2: Step-level-only is not optimal.** From Lemma 2, we can know that $f_{async}(\gamma_1)$ is strictly increasing when $1 \leq \gamma \leq \lceil \frac{1}{c_1} \rceil$, and will stay constant after that.

**1.** $M \leq \lceil \frac{1}{c_1} \rceil$ This case, according to Lemma 1,

$$h(\gamma_1, 1) \leq h(M, 1) < h(\frac{M}{2}, 2), \forall \gamma_1 \leq M$$

So step-level-only is not optimal.

**2.**$\lceil \frac{1}{c_1} \rceil < M < 2\lceil \frac{1}{c_1} \rceil$ This case, according to Lemma 1

$$h(\gamma_1, 1) \leq h(\lceil \frac{1}{c_1} \rceil, 1) = h(M, 1) < h(\frac{M}{2}, 2), \forall \gamma_1 \leq M$$

**3.** $M \geq 2\lceil \frac{1}{c_1} \rceil$ This case, we will have

$$h(\gamma_1, 1) \leq h(\lceil \frac{1}{c_1} \rceil, 1) < h(\lceil \frac{1}{c_1} \rceil, 2), \forall 1 \leq \gamma_1 \leq M$$

So step-level-only is not optimal.

$\square$

**Lemma 1.** *Let*
$$w(\gamma) = f(\gamma)\, g(M/\gamma),$$
*where M is an even number, and*

- $f(\gamma) = \dfrac{1 - \alpha_1^\gamma}{1 - \alpha_1 + c_1\, \alpha_1(1 - \alpha_1^\gamma) - c_1\, \gamma\, \alpha_1^\gamma(1 - \alpha_1)}$,

- $g(\gamma) = \dfrac{1 - \alpha_2^{\gamma}}{(1 - \alpha_2)(1 - c_2 + c_2\,\gamma)},$

- $0.5 < \alpha_1, \alpha_2 < 0.8, 0 < c_1 < \frac{1}{3}, 0 < c_2 < \frac{1}{5}$ and $M \geq 16$.

*Then $w(2) > w(1)$, $w(\frac{M}{2}) > w(M)$.*

*Proof.*

$$w(2)/w(1) = \frac{(1 + \alpha_1)}{(1 + \alpha_2^{M/2})(1 + \alpha_1 c_1(1 - \alpha_1))}\frac{1 - c_2 + c_2 M}{1 - c_2 + c_2 M/2}$$

$$> \frac{(1 + \alpha_1)}{(1 + \alpha_2^{M/2})(1 + \alpha_1 c_1(1 - \alpha_1))} \times 1$$

We can easily found that this function is monotonically decreasing with respect to $c_1$, $\alpha_2$ $M$, monotonically increasing with respect to $\alpha_1$. Therefore,

$$w(2)/w(1) > \frac{1 + 0.52}{(1 + 0.8^8)(1 + 0.52 \times 1/3 \times 0.48)} > 1$$

$$w(\frac{M}{2})/w(M) = \frac{1 + \alpha_2}{1 + c_2}\frac{1 - \alpha_1 + c_1\alpha_1(1 - \alpha_1^M) - c_1 M\alpha_1^M(1 - \alpha_1)}{(1 + \alpha_1^{M/2})(1 - \alpha_1 + c_1\alpha_1(1 - \alpha_1^{M/2}) - c_1\frac{M}{2}\alpha_1^{M/2}(1 - \alpha_1))}$$

$$= \frac{1 + \alpha_2}{1 + c_2}[1 - (1 - c_1 M\frac{1 - \alpha_1^{M/2}}{2})\frac{\alpha_1^{M/2}}{1 + \alpha_1^{M/2}}\frac{1 - \alpha_1}{(1 - \alpha_1)(1 + c_1(\alpha_1 + \cdots + \alpha_1^{M/2}) - c_1\frac{M}{2}\alpha_1^{M/2})}]$$

Given that $\alpha_1^k \geq \alpha_1^{M/2}, k \in \{1, 2, \cdots, M/2\}$, so we have

$$c_1(\alpha_1 + \alpha_1^2 + \cdots + \alpha_1^{M/2}) - c_1\frac{M}{2}\alpha_1^{M/2} > 0$$

If

$$1 - c_1 M\frac{1 - \alpha_1^{M/2}}{2} \leq 0$$

then

$$w(\frac{M}{2})/w(M) \geq \frac{1 + \alpha_2}{1 + c_2} > 1$$

Otherwise,

$$w(\frac{M}{2})/w(M) > \frac{1 + \alpha_2}{1 + c_2}[1 - 1 \times \frac{\alpha_1^{M/2}}{1 + \alpha_1^{M/2}} \times 1] > \frac{1 + 0.52}{1 + 1/5}(1 - \frac{0.8^8}{1 + 0.8^8}) > 1$$

$\square$

**Lemma 2.** *The speedup function of the async* LOOKAHEAD REASONING

$$f_{async}(\gamma) = \begin{cases} S_2 = \dfrac{1 - \alpha_1^{\gamma}}{1 - \alpha_1 + c_1\,\alpha_1\,(1 - \alpha_1^{\gamma}) - c_1\,\gamma\,\alpha_1^{\gamma}\,(1 - \alpha_1)}, & \gamma < \lceil \frac{1}{c_1} \rceil \\[3mm] S_1 = \dfrac{1}{c_1 + (1 - c_1)(1 - \alpha_1)}, & otherwise \end{cases}$$

*is strictly increasing when $1 \leq \gamma \leq \lceil \frac{1}{c_1} \rceil, \gamma \in \mathbb{N}^+$, and will stay constant after $\gamma \geq \lceil \frac{1}{c_1} \rceil, \gamma \in \mathbb{N}^+$*

*Proof.* **We first see the function as a continuous function on $\mathbb{R}$ and prove that when $1 \leq \gamma < \lceil \frac{1}{c_1} \rceil$, the speedup function is strictly increasing**

Write
$$A(\gamma) = 1 - \alpha_1^\gamma, \qquad D(\gamma) = 1 - \alpha_1 + c_1\,\alpha_1\,(1 - \alpha_1^\gamma) - c_1\,\gamma\,\alpha_1^\gamma\,(1 - \alpha_1).$$
Then $f_{async}(\gamma) = A(\gamma)/D(\gamma)$, and by the quotient rule
$$f'_{async}(\gamma) = \frac{A'(\gamma)\,D(\gamma)\;-\;A(\gamma)\,D'(\gamma)}{D(\gamma)^2}.$$

We compute
$$A'(\gamma) = -\alpha_1^\gamma \ln \alpha_1, \qquad D'(\gamma) = -c_1\,\alpha_1^\gamma\Big[\alpha_1 \ln \alpha_1 + (1 - \alpha_1)\big(1 + \gamma \ln \alpha_1\big)\Big].$$

Hence the numerator of $f'_{async}(\gamma)$ becomes

$$
\begin{aligned}
A'(\gamma)\,D(\gamma)\;-\;A(\gamma)\,D'(\gamma) &= \alpha_1^\gamma\Big\{(-\ln \alpha_1)\,D(\gamma) + c_1\,(1 - \alpha_1^\gamma)\Big[\alpha_1 \ln \alpha + (1 - \alpha_1)(1 + \gamma \ln \alpha_1)\Big]\Big\} \\
&= \alpha_1^\gamma(1 - \alpha_1)\Big[(-\ln \alpha_1) + c_1\,\gamma\,\alpha_1^\gamma\,(\ln \alpha_1) + c_1\,(1 - \alpha_1^\gamma)\,(1 + \gamma \ln \alpha_1)\Big] \\
&= \alpha_1^\gamma(1 - \alpha_1)\Big[(-\ln \alpha_1)(1 - c_1\gamma) + c_1(1 - \alpha_1^\gamma)\Big]
\end{aligned}
$$

Since
$$-\ln \alpha_1 > 0, \quad 1 - \alpha_1 > 0, \quad 1 - c_1\gamma > 0, \quad 1 - \alpha_1^\gamma > 0$$
each term is strictly positive for all $\gamma \geq 1$. Therefore
$$A'(\gamma)\,D(\gamma) - A(\gamma)\,D'(\gamma) > 0 \quad \Longrightarrow \quad f'_{async}(\gamma) > 0,$$

**Then we prove that** $S_1 \geq S_2$ We only need to prove that when $\gamma = \frac{1}{c_1}$, we have $S_2 = S_1$

When $\gamma = \frac{1}{c_1}$,

$$
\begin{aligned}
S_2 &= \frac{1 - \alpha_1^\gamma}{1 - \alpha_1 + c_1\,\alpha_1\,(1 - \alpha_1^\gamma) - c_1\,\gamma\,\alpha_1^\gamma\,(1 - \alpha_1)} \\
&= \frac{1 - \alpha_1^\gamma}{1 - \alpha_1 + c_1\,\alpha_1\,(1 - \alpha_1^\gamma) - \alpha_1^\gamma\,(1 - \alpha_1)} \\
&= \frac{1 - \alpha_1^\gamma}{(1 - \alpha_1)(1 - \alpha_1^\gamma) + c_1\,\alpha_1\,(1 - \alpha_1^\gamma)} = \frac{1}{c_1 + (1 - c_1)(1 - \alpha_1)} = S_1
\end{aligned}
$$

Therefore, So the lemma follows. $\qquad\square$

**Lemma 3.** *The speedup function of token-level speculative decoding[3] and is a unimodality function.*

$$g(\gamma) = \frac{1 - \alpha_2^\gamma}{(1 - \alpha_2)\,(1 - c_2 + c_2\,\gamma)}, \gamma \in \mathbb{R}$$

*where $\alpha_2 > c_2$ Then $g$ increases on $[1, \hat{\gamma})$ and decreases on $(\hat{\gamma}, \infty)$ for a unique $\hat{\gamma} \in \mathbb{R}^+$. In practical scenarios, where $\gamma \in \mathbb{N}^+$, the maximum of $g(\gamma)$ is attained at some integer point $\gamma^* \in \mathbb{N}^+, \gamma^* \geq 2$ Sync* LOOKAHEAD REASONING *has a similar form, so it also has this property.*

*Proof.* **Step 1. Compute** $g'(\gamma)$**.** Set
$$N(\gamma) = 1 - \alpha_2^\gamma, \qquad D(\gamma) = (1 - \alpha_2)\big(1 - c_2 + c_2\,\gamma\big),$$
so that $g(\gamma) = N(\gamma)/D(\gamma)$. By the quotient rule,
$$g'(\gamma) = \frac{N'(\gamma)\,D(\gamma) - N(\gamma)\,D'(\gamma)}{D(\gamma)^2}.$$

Since
$$N'(\gamma) = -\alpha_2^\gamma \ln \alpha_2, \qquad D'(\gamma) = (1 - \alpha_2)\,c_2,$$
$$g'(\gamma) = \frac{\alpha_2^\gamma\,(-\ln \alpha_2)\,(1 - \alpha_2)\,(1 - c_2 + c_2\,\gamma)\;-\;c_2\,(1 - \alpha_2^\gamma)\,(1 - \alpha_2)}{[(1 - \alpha_2)\,(1 - c_2 + c_2\,\gamma)]^2}.$$

Since $(1 - \alpha_2) > 0$, the sign of $g'(\gamma)$ equals the sign of

$$F(\gamma) := \alpha_2^\gamma \left(- \ln \alpha_2\right) \left(1 - c_2 + c_2\, \gamma\right) \; - \; c_2 \left(1 - \alpha_2^\gamma\right).$$

**Step 2.** $F$ **is strictly decreasing.** Differentiate $F$:

$$F'(\gamma) = -\alpha_2^\gamma \left(\ln \alpha_2\right)^2 \left(1 - c_2 + c_2\, \gamma\right).$$

Since $1 - c_2 + c_2\gamma > 0$, it follows

$$F'(\gamma) < 0 \quad \text{for all } \gamma \geq 1.$$

Thus $F$ is strictly decreasing on $[1, \infty)$.

**Step 3. Signs of $F$ at the ends.**

- At $\gamma = 1$:
$$F(1) = \alpha_2(- \ln \alpha_2) - c_2(1 - \alpha_2)$$
since $\alpha_2 > c_2 > 0, - \ln \alpha_2 > 1 - \alpha_2 > 0$, so we have $F(1) > 0$

- As $\gamma \to \infty$, $\alpha_2^\gamma \to 0$, so
$$F(\gamma) = \alpha_2^\gamma \left[(- \ln \alpha)\left(1 - c_2 + c_2\, \gamma\right) + c\right] - c_2 \left(1 - \alpha_2^\gamma\right) \; \longrightarrow \; -c < 0.$$

**Step 4. Conclusion via the Intermediate Value Theorem.** Because $F$ is continuous, strictly decreasing, $F(1) > 0$, and $\lim_{\gamma \to \infty} F(\gamma) < 0$, there exists a *unique* $\hat{\gamma} \in \mathbb{R}, \hat{\gamma} \geq 1$ such that $F(\hat{\gamma}) = 0$. Moreover,

$$F(\gamma) > 0 \quad \Leftrightarrow \quad 1 \leq \gamma < \hat{\gamma}, \qquad F(\gamma) < 0 \quad \Leftrightarrow \quad \gamma > \hat{\gamma}.$$

Since $g'(\gamma)$ and $F(\gamma))$ share the same sign, it follows that

$$g'(\gamma) > 0 \text{ for } 1 \leq \gamma < \hat{\gamma}, \qquad g'(\gamma) < 0 \text{ for } \gamma > \hat{\gamma},$$

Besides, noting that

$$g(2)/g(1) = \frac{1 + \alpha_2}{1 + c_2}$$

so in practical scenarios where $\gamma \in \mathbb{N}^+$, we conclude that the maximum is achieved at some integer point $\gamma^* \geq 2$ as claimed. $\qquad \square$

**Lemma 4.** *Let* $0.5 < \alpha < 0.8$ *and define*

$$a(\alpha, x) \; = \; - \ln \alpha \; \frac{x\, \alpha^x}{1 - \alpha^x}, \qquad x \geq 1.$$

*Then:*

1. *For each fixed* $\alpha \in (0.5, 0.8)$*, the function* $x \mapsto a(\alpha, x)$ *is strictly decreasing on* $[1, \infty)$*.*

2. *For each fixed* $x \geq 1$*, the function* $\alpha \mapsto a(\alpha, x)$ *is strictly increasing on* $(0.5, 0.8)$*.*

3. *Consequently, for every* $x \geq 10$ *and* $\alpha \in (0.5, 0.8)$*,*

$$a(\alpha, x) \; < \; a(0.8, 10) \; = \; - \ln(0.8)\, \frac{10 \cdot 0.8^{10}}{1 - 0.8^{10}} \; \approx \; 0.26,$$

*and for all* $\alpha \in (0.52, 0.8)$*,*

$$a(\alpha, 2) \; \in \; \big(a(0.52, 2),\, a(0.8, 2)\big) \; \approx \; (0.48,\; 0.79).$$

*Proof.* **(i) Monotonicity in $x$.** Fix $\alpha \in (0,1)$ and write

$$f(x) = \frac{x\,\alpha^x}{1 - \alpha^x} = \frac{N(x)}{D(x)}, \quad N(x) = x\,\alpha^x, \quad D(x) = 1 - \alpha^x.$$

Then

$$N'(x) = \alpha^x\big(1 + x\ln\alpha\big), \qquad D'(x) = -\alpha^x \ln\alpha,$$

and by the quotient rule

$$f'(x) = \frac{N'(x)\,D(x) - N(x)\,D'(x)}{D(x)^2} = \frac{\alpha^x\big[(1 + x\ln\alpha)(1 - \alpha^x) - x(-\ln\alpha)\,\alpha^x\big]}{(1 - \alpha^x)^2}.$$

Since $0 < \alpha < 1$, setting $u = x\ln\alpha < 0$ we have by convexity of the exponential,

$$\alpha^x = e^u > 1 + u = 1 + x\ln\alpha,$$

hence

$$(1 + x\ln\alpha)(1 - \alpha^x) - x(-\ln\alpha)\,\alpha^x = 1 + x\ln\alpha - \alpha^x < 0.$$

Thus $f'(x) < 0$. Because $-\ln\alpha > 0$, it follows immediately

$$\frac{\partial}{\partial x}\, a(\alpha, x) = -\ln\alpha\ f'(x) < 0,$$

so $a(\alpha, x)$ is strictly decreasing in $x \geq 1$.

**(ii) Monotonicity in $\alpha$.** Fix $x \geq 1$ and set

$$U(\alpha) = -\ln\alpha, \qquad V(\alpha) = \frac{x\,\alpha^x}{1 - \alpha^x},$$

so $a(\alpha, x) = U(\alpha)\,V(\alpha)$. Then

$$U'(\alpha) = -\frac{1}{\alpha}, \quad V'(\alpha) = \frac{x^2\,\alpha^{x-1}}{(1 - \alpha^x)^2} > 0.$$

Hence

$$\frac{\partial a}{\partial \alpha} = U'(\alpha)\,V(\alpha) + U(\alpha)\,V'(\alpha) = -\frac{V(\alpha)}{\alpha} + (-\ln\alpha)\,V'(\alpha).$$

We claim this is $> 0$. Indeed,

$$-\frac{V}{\alpha} + (-\ln\alpha)\,V' > 0 \quad \Longleftrightarrow \quad (-\ln\alpha)\,\alpha\,V' > V.$$

Since

$$\alpha\,V' = \alpha\,\frac{x^2\alpha^{x-1}}{(1 - \alpha^x)^2} = \frac{x^2\,\alpha^x}{(1 - \alpha^x)^2}, \quad V = \frac{x\,\alpha^x}{1 - \alpha^x},$$

this inequality becomes

$$(-\ln\alpha)\,\frac{x^2\,\alpha^x}{(1 - \alpha^x)^2} > \frac{x\,\alpha^x}{1 - \alpha^x} \quad \Longleftrightarrow \quad x\,(-\ln\alpha) > 1 - \alpha^x.$$

But for $0.5 < \alpha < 0.8$, the well-known bound $-\ln\alpha > 1 - \alpha$ and $\alpha^x \leq \alpha$ imply

$$x\,(-\ln\alpha) \;\geq\; -\ln\alpha \;>\; 1 - \alpha \;\geq\; 1 - \alpha^x,$$

so $\partial a/\partial\alpha > 0$. Thus $a(\alpha, x)$ is strictly increasing in $\alpha \in (0.5, 0.8)$.

**(iii) Numerical bounds.** By (i), $a(\alpha, x) \leq a(\alpha, 10)$ for all $x \geq 10$, and by (ii),

$$a(\alpha, 10) \leq a(0.8, 10) = -\ln(0.8)\,\frac{10 \cdot 0.8^{10}}{1 - 0.8^{10}} \approx 0.26.$$

$$a(\alpha, 6) \leq a(0.8, 6) = -\ln(0.8)\,\frac{10 \cdot 0.8^6}{1 - 0.8^6} \approx 0.48.$$

$$a(\alpha, 8) \leq a(0.8, 8) = -\ln(0.8)\,\frac{10 \cdot 0.8^8}{1 - 0.8^8} \approx 0.36.$$

Also by (ii), for any $\alpha \in (0.52, 0.8)$,

$$a(\alpha, 2) \;\in\; \big(a(0.52, 2),\, a(0.8, 2)\big) \in (0.48, 0.8).$$

This completes the proof. $\qquad\square$

**Lemma 5.** *Let*
$$w(\gamma) = f_{sync}(\gamma)\, g(M/\gamma),$$
*here* $\gamma \in \mathbb{N}^+, M/\gamma \in \mathbb{N}^+$ *and assume* $M \geq 4$. *Then:*

(a) *If both (8) and (9) hold, then* $w(\gamma)$ *is unimodal on* $[1, M]$ *and attains its maximum at some* $\gamma^* \in [2, \frac{M}{2}]$.

(b) *If (8) fails, then the unique maximizer is* $\gamma = 1$ *(token-level only).i.e.*
$$h(\gamma_1, \gamma_2) < h(1, M), \forall \gamma_1 \gamma_2 = M, \gamma_1, \gamma_2 \in \mathbb{N}^+$$

(c) *If (9) fails, then unique maximizer is* $\gamma = M$ *(step-level only). i.e.*
$$h(\gamma_1, \gamma_2) < h(M, 1), \forall \gamma_1 \gamma_2 = M, \gamma_1, \gamma_2 \in \mathbb{N}^+$$

*Proof.* We first treat this function as a continuous function over $\mathbb{R}$, analyze its derivative to determine its monotonicity, and then restrict its domain to $\mathbb{N}^+$ to obtain the desired results.

**Step 1: Derivative.** By the product and chain rules,
$$w'(\gamma) = \frac{1}{\gamma}\, f_{sync}(\gamma)\, g(M/\gamma) \left[\gamma\, \frac{f'_{sync}(\gamma)}{f_{sync}(\gamma)} - \frac{M}{\gamma}\, \frac{g'(M/\gamma)}{g(M/\gamma)}\right].$$

**Step 2: Log-derivatives.** Define
$$a_i(x) = x\, \frac{-\ln(\alpha_i)\, \alpha_i^x}{1 - \alpha_i^x}, \quad i = 1, 2.$$
Then
$$\gamma \frac{f'_{sync}(\gamma)}{f_{sync}(\gamma)} = a_1(\gamma) - \frac{c_1 \gamma}{1 - c_1 + c_1 \gamma}, \quad \gamma \frac{g'(\gamma)}{g(\gamma)} = a_2(\gamma) - \frac{c_2 \gamma}{1 - c_2 + c_2 \gamma}.$$
Hence with $\eta = M/\gamma$,
$$w'(\gamma) = \frac{f_{sync}(\gamma)\, g(\eta)}{\gamma} \left[a_1(\gamma) - a_2(\eta) - \left(\frac{c_1 \gamma}{1 - c_1 + c_1 \gamma} - \frac{c_2 \eta}{1 - c_2 + c_2 \eta}\right)\right].$$

**Step 3: Monotonicity.** By Lemma 4, $a_1(\gamma)$ is strictly decreasing in $\gamma$ and $a_2(M/\gamma)$ strictly increasing. Also
$$\frac{c_1 \gamma}{1 - c_1 + c_1 \gamma} - \frac{c_2 \eta}{1 - c_2 + c_2 \eta} = 1 - \frac{1 - c_1}{1 - c_1 + c_1 \gamma} - \frac{c_2 M}{(1 - c_2)\gamma + c_2 M}$$
is strictly increasing in $\gamma$. Therefore $w'(\gamma)$ is strictly decreasing on $[1, M]$, and so $w(\gamma)$ either strictly increasing, or strictly decreasing, or first increasing then decreasing on $[1, M]$.

**Step 4: Endpoint comparison.** Now we restrict the domain to be $\mathbb{N}^+$

- If (8) fails, then $w(2) < w(1)$, so $w$ is either strictly decreasing on $[1, M]$, or first increasing then decreasing and the critical points was in $(1, 2)$. Therefore we can conclude that $w$ is strictly decreasing on $[2, M]$. Therefore we can conclude that there's unique maximize at $\gamma = 1$, so we have
$$h(\gamma_1, \gamma_2) < h(1, M), \gamma_1 \gamma_2 = M, \gamma_1, \gamma_2 \in \mathbb{N}^+$$

- If (9) fails, then $w(\frac{M}{2}) < w(M)$, so $w$ is either strictly increasing on $[1, M]$, or first increasing then decreasing and the critical points was in $(\frac{M}{2}, M)$. Therefore we can conclude that $w$ is strictly increasing on $[1, \frac{M}{2}]$. Therefore we can conclude that there's unique maximize at $\gamma = M$, so we have
$$h(\gamma_1, \gamma_2) < h(M, 1), \gamma_1 \gamma_2 = M, \gamma_1, \gamma_2 \in \mathbb{N}^+$$

- If both (8) and (9) holds, then $w(2) \geq w(1), w(\frac{M}{2}) \geq w(M)$, so $w$ would achieve the max between $[2, \frac{M}{2}]$

So the lemma follows. □

**Lemma 6.** *When* $0 < y < 1$

$$F(y) = 2 - y + (y - 2 + \frac{1}{y}) \ln(1 - y) < 1.157$$

*Proof.* Define

$$F(y) = 2 - y + (y - 2 + y^{-1}) \ln(1 - y), \qquad 0 < y < 1.$$

**First derivative.** A direct calculation gives

$$F'(y) = \frac{d}{dy}\left[(y - 2 + y^{-1}) \ln(1 - y)\right] - 1 = (1 - y^{-2}) \ln(1 - y) - \frac{1}{y}.$$

**Second derivative and concavity of $F'$.** Differentiating again,

$$F''(y) = \frac{d}{dy}\left[(1 - y^{-2}) \ln(1 - y)\right] + \frac{d}{dy}(-y^{-1})$$

$$= \frac{2 \ln(1 - y)}{y^3} - \frac{1 - y^{-2}}{1 - y} + \frac{1}{y^2} = \frac{y(1 + y) + 2 \ln(1 - y)}{y^3}.$$

Set

$$N(y) = y(1 + y) + 2 \ln(1 - y),$$

so that $F''(y) = N(y)/y^3$. On $(0, 1)$,

$$N'(y) = \frac{d}{dy}\left[y + y^2 + 2 \ln(1 - y)\right] = 1 + 2y - \frac{2}{1 - y} = -\frac{2y^2 - y + 1}{1 - y} < 0,$$

and $N(0) = 0$. Hence $N(y) < 0$ for all $y \in (0, 1)$, which implies

$$F''(y) < 0 \quad \text{on } (0, 1).$$

Thus $F'$ is strictly decreasing.

**Sign-change of $F'$.** - As $y \to 0^+$, $\ln(1 - y) \sim -y$, so

$$F'(y) \sim (-y)(1 - y^{-2}) - \tfrac{1}{y} = +O(y) > 0.$$

- As $y \to 1^-$, $\ln(1 - y) \to -\infty$ while $(1 - 1/y^2) \to 1$, so $F'(y) \to -\infty$.

By continuity and strict decrease, there is a unique $y^* \in (0, 1)$ with $F'(y^*) = 0$, and

$$F'(y) > 0 \quad (0 < y < y^*), \qquad F'(y) < 0 \quad (y^* < y < 1).$$

Hence $F$ increases on $(0, y^*)$ and decreases on $(y^*, 1)$, so its maximum on $(0, 1)$ occurs at $y^*$.

**Numerical evaluation.** Numerically one finds

$$y^* \approx 0.5693971022, \qquad F(y^*) \approx 1.1562281731 < 1.157.$$

**Conclusion.** Therefore for all $y \in (0, 1)$,

$$F(y) \leq F(y^*) < 1.157,$$

as claimed. □

# C   Calculation of $S_2(n)$

We'll prove when $n \to \infty$

$$S_2(n) = \frac{n + \sum_{i=1}^{n} X_i}{\sum_{i=1}^{n} \lceil \frac{X_i + 1}{\gamma} \rceil + c_1 \sum_{i=1}^{n} (X_i \bmod \gamma)} \longrightarrow \frac{1 - \alpha_1^{\gamma}}{(1 - \alpha)_1 + c_1 \left[\alpha_1 - \alpha_1^{\gamma+1} - \gamma(1 - \alpha_1)\alpha_1^{\gamma}\right]}.$$

**Step 1: Use the Law of Large Numbers** According to the Law of Large Numbers

$$S_2(n) = \frac{1 + \frac{1}{n}\sum_{i=1}^{n} X_i}{\frac{1}{n}\sum_{i=1}^{n}\left\lceil\frac{X_i+1}{\gamma}\right\rceil + \frac{c_1}{n}\sum_{i=1}^{n}(X_i \bmod \gamma)} \longrightarrow \frac{1 + E[X_i]}{E\left\lceil\frac{X_i+1}{\gamma}\right\rceil + c_1 E[X_i \bmod \gamma]}$$

Here, the PMF of $X_i$ is:

$$P(X_i = k) = \alpha_1^k(1 - \alpha_1)$$

And its expectation is:

$$E[X_i] = \frac{\alpha_1}{1 - \alpha_1}$$

**Step 2: Compute** $E\left[\left\lceil\frac{X_i+1}{\gamma}\right\rceil\right]$ Let $Y = \left\lceil\frac{X_i+1}{\gamma}\right\rceil$.

**Key Observation** The ceiling function $\left\lceil\frac{X_i+1}{\gamma}\right\rceil$ can be expressed in terms of integer thresholds. For $m \geq 0$:

$$\left\lceil\frac{X_i + 1}{\gamma}\right\rceil = m + 1 \quad \text{if} \quad X_i \in [m\gamma, (m+1)\gamma - 1]$$

Thus:

$$Y = m + 1 \quad \text{for} \quad X_i \in [m\gamma, (m+1)\gamma - 1], \quad m = 0, 1, 2, \ldots$$

**Compute** $E[Y]$

$$E[Y] = \sum_{m=0}^{\infty}(m+1)P\left(m\gamma \leq X_i \leq (m+1)\gamma - 1\right)$$

The probability $P\left(m\gamma \leq X_i \leq (m+1)\gamma - 1\right)$ is:

$$\sum_{k=m\gamma}^{(m+1)\gamma-1} P(X_i = k) = \alpha_1^{m\gamma}(1 - \alpha_1^{\gamma})$$

Therefore we can get:

$$E\left[\left\lceil\frac{X_i + 1}{\gamma}\right\rceil\right] = \frac{1}{1 - \alpha_1^{\gamma}}$$

**Step 3: Compute** $E[X_i \bmod \gamma]$ To calculate the expectation of $X_i \bmod \gamma$, where $X_i$ follows the given geometric distribution and $\gamma$ is an integer greater than 4, we proceed as follows:

**Compute** $P(X_i \bmod \gamma = r)$ For $r \in \{0, 1, \ldots, \gamma - 1\}$, we have:

$$P(X_i \bmod \gamma = r) = \sum_{m=0}^{\infty} P(X_i = m\gamma + r) = \frac{(1 - \alpha_1)\alpha_1^r}{1 - \alpha_1^{\gamma}}$$

**Compute** $E[X_i \bmod \gamma]$ The expectation is:

$$E[X_i \bmod \gamma] = \sum_{r=0}^{\gamma-1} r \cdot P(X_i \bmod \gamma = r) = \sum_{r=0}^{\gamma-1} r \cdot \frac{(1 - \alpha_1)\alpha_1^r}{1 - \alpha_1^{\gamma}} = \frac{\alpha_1 - \alpha_1^{\gamma+1} - \gamma(1 - \alpha_1)\alpha_1^{\gamma}}{(1 - \alpha_1)(1 - \alpha_1^{\gamma})}$$

# D Illustration of Hybrid Approach

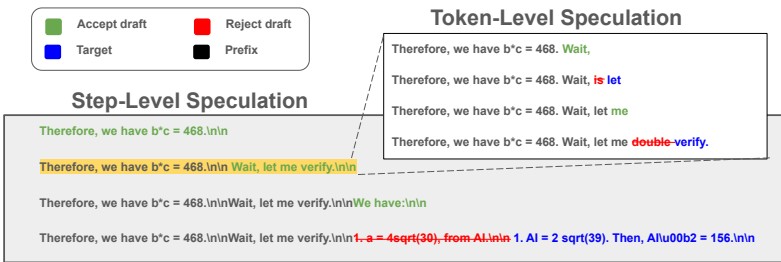

Figure 4: Illustration of hybrid approach.

# E  Verifier Comparison on QwQ-32B for Direct SPECREASON Reproduction

To facilitate a fair comparison with SPECREASON, we replicate their setup using QwQ-32B as the base model and deepseek-r1-distill-1.5B as the draft model, which is the original model setting in SPECREASON. Table 4 summarizes the results across multiple verifier strategies, reporting accuracy (%) and acceptance rate (%) on GSM8K and AIME. All values are averaged over 16 runs.

| Dataset | Base | Rand. | LLM-J 32B | LLM-J 7B | Emb 0.95 | Emb 0.85 | Score 0.7 | Score 0.8 |
|---|---|---|---|---|---|---|---|---|
| GSM8K | 96.2 | 96.5 (40) | 96.3 (43) | 96.4 (50) | 96.4 (17) | 96.2 (38) | 95.6 (76) | 95.4 (36) |
| AIME | 77.1 | 75.8 (30) | 77.5 (38) | 75.0 (42) | 77.7 (24) | 74.4 (36) | 66.9 (41) | 74.6 (16) |

Table 4: Accuracy and acceptance rate (%) of QwQ-32B under different verifier strategies. Each cell shows: Accuracy (Acceptance Rate).

As shown above, both the LLM-as-a-Judge (LLM-J) verifier and the embedding-based verifier with a high similarity threshold (0.95) maintain the baseline accuracy on AIME (77.5% and 77.7% respectively, compared to 77.1% without verification), while the score-based verifiers used in SPECREASON result in notable degradation, dropping to as low as 66.9% when using the score threshold of 0.7. These findings emphasize that scoring drafts with the same model does not align well with its own generation distribution, particularly under longer contexts. Our experiments use the full 32k context length, whereas the original SPECREASON paper reported results using a truncated 8k context. This difference further amplifies the mismatch between score-based filtering and semantic correctness, especially on complex tasks like AIME.

# F  Verifier Latency and Speedup Analysis on QwQ-32B

To analyze the accuracy-efficiency trade-off, we report the end-to-end inference speedups for various verifiers on QwQ-32B model with the deepseek-r1-distill-1.5B draft model. These experiments were conducted on an H200 server (with TP=2 for the 32B verifier) and include results with n-gram speculation to show orthogonality.

| Dataset | LLM-J (32B) | LLM-J (32B)+n-gram | LLM-J (7B) | LLM-J (7B)+n-gram | Emb (0.85) | Emb (0.85)+n-gram | Score (0.7) | Score (0.7)+n-gram |
|---|---|---|---|---|---|---|---|---|
| GSM8K | 1.32× | 1.67× | 1.45× | 1.75× | 1.22× | 1.53× | 1.57× | 1.63× |
| AIME | 1.12× | 1.42× | 1.20× | 1.57× | 1.10× | 1.41× | 0.77× | 0.93× |

Table 5: End-to-end inference speedup (× faster) of different verifiers with and without n-gram speculation on QwQ-32B. Experiments run on H200 (TP=2 for 32B).

The results reveal key trade-offs between verifier latency, acceptance rate, and final speedup. For instance, the 7B LLM-J achieves greater overall acceleration than both the heavier 32B LLM-J and the more lightweight Embedding model. This is particularly interesting given their isolated per-step latencies: the Embedding model is fastest (0.009s), followed by the 7B LLM-J (0.013s), and the 32B LLM-J (0.025s). We hypothesize that the final speedup is a function of both low latency and high acceptance rate, and the 7B verifier strikes the most effective balance between these two factors.

We analyzed the performance of the SpecReason verifier (Score 0.7). While it achieves a significant 1.57x speedup on the short-context GSM8K dataset (avg. output 1K tokens), we observed an overall slowdown (0.77x) on the long-context AIME dataset. Upon analyzing the generation traces, we found that SpecReason actually causes a deceleration on individual examples where the output length grows very long (often >10K tokens). We hypothesize this slowdown stems from a lack of system optimizations in SpecReason, where per-step costs like tokenizer overhead become amplified for these long sequences. This finding is consistent with the original SpecReason paper, where experiments on shorter contexts (<8k tokens) would not have revealed this overhead. We also note that for practical applications of large reasoning models, a context limit of 8k tokens can be overly restrictive.

Finally, the +n-gram results consistently show a substantial boost across all methods, confirming that both our LOOKAHEAD REASONING and SpecReason are orthogonal to token-level speculation.

