# OpenReview forum: "Scaling Speculative Decoding with Lookahead Reasoning"
_NeurIPS.cc/2025/Conference — NeurIPS 2025 poster_

### Official Review · Reviewer_WAjf · 2025-06-04

**Clarity:** 4
**Significance:** 3
**Originality:** 4
**Rating:** 4
**Confidence:** 4

**Summary:**

This work proposes Lookahead Reasoning that uses a smaller draft model to propose reasoning steps for a larger reasoning models. A third model is used to evaluate whether the drafted steps should be accepted based on semantic similarity. When used together with token level speculative decoding, generation efficiency can be improved without losing too much accuracy.

**Questions:**

1. SpecReason uses QwQ-32B models as the base model and reported a much lower accuracy on AIME for the original model and using reasoning model scoring actually improved model accuracy for them. Is it possible to get some results (i.e. Table 2) using the same model with different verifiers so that the comparison is straight-forward?
2. (Related to weakness 1) Can you report the speed-up of SpecReason and SpecReason+SD?
3. (Related to weakness 2) Can you report the latency (or end to end speed-up) of using different verifiers?

**Ethical Concerns:**

["NO or VERY MINOR ethics concerns only"]

**Final Justification:**

The author provided experiment results to complete the evaluation. Therefore, I raise my rating.

**Limitations:**

1. Auto-regressive calls to the base model: If I understand the paper correctly, this work still involves auto-regressively generating from the large base model. This means that there is opportunity to gain further speed-up if we are able to cut down the amount of auto-regressive generation from the target model.
2. Token level speculation: The reviewer understands that the method proposed is orthogonal to token level speculation. The token level speculation method (prompt look-up decoding) used in the experiment is far from SOTA. While not discrediting this work, it would be nice if we can see more recent speculative decoding methods used together with LR and their combined effect on efficiency.

**Quality:**

3

**Strengths And Weaknesses:**

Strength:
1. Presentation is clear and understandable
2. Experiments results are good. However, some key comparisons are missing.

Weakness:
1. Key comparison with baseline: The work mentioned and compared against SpecReason as a baseline. However, the speed-up obtained using SpecReason is not reported in Table1. Both methods involves a trade-off between end-to-end speed up (efficiency) and results accuracy. Comparing only with the results accuracy with your baseline seems unfair.  Meanwhile, SpecReason is also orthogonal to token-level speculation. Hence a complete comparison should be SpecReason+SD (Ngram) with LR+SD (Ngram) in terms of both speed-up and accuracy.
2. Efficiency of different verifiers: Different verifiers has different runtime and it is insufficient to report the acceptance rate in Table 2 when comparing different verifiers. Verifier runtime (or even end-to-end runtime) should be reported in order for the readers to evaluate the trade-off more comprehensively.
3. The methods employs an additional 7B model as a verifier to evaluate whether to accept the drafted steps. This puts a lot of pressure on the memory and adds a lot of latency to the critical path of execution.

---

> ### Author Rebuttal · Authors · 2025-07-31
>
> Thank you for your thoughtful review and the encouraging feedback. Below, we address your questions in detail.
>
> **Q1:** SpecReason uses QwQ-32B models as the base model and reported a much lower accuracy on AIME for the original model and using reasoning model scoring actually improved model accuracy for them. Is it possible to get some results (i.e. Table 2) using the same model with different verifiers so that the comparison is straight-forward?
>
> **A1:** For a direct comparison with SpecReason, we evaluated the QwQ-32B model with the deepseek-r1-distill-1.5B draft model, as used in their paper. The results below show that verifiers grounded in semantic equivalence are most effective at preserving the model's original accuracy.
>
> **Performance of QwQ-32B with Various Verifiers**
> *Each cell shows: Accuracy% (Acceptance Rate %), Accuracy values are averaged over 16 runs.*
> | Dataset | QwQ-32B | Rand. | LLM J.(32B) | LLM J(7B) | Emb (0.95) | Emb (0.85) | Score (0.7) | Score (0.8) |
> |---------|---------|-------|-------------|-----------|------------|------------|-------------|-------------|
> | GSM8K   | 96.2%   | 96.5% (40%) | 96.3% (43%) | 96.4% (50%) | 96.4% (17%) | 96.2% (38%) | 95.6% (76%) | 95.4% (36%) |
> | AIME    | 77.1%   | 75.8% (30%) | 77.5% (38%) | 75.0% (42%) | 77.7% (24%) | 74.4% (36%) | 66.9% (41%) | 74.6% (16%) |
>
>
> Our LLM-as-a-Judge (LLM-J) verifier and the Embedding model with a high similarity threshold (0.95) both successfully maintained the baseline accuracy on the challenging AIME dataset (77.5% and 77.7% vs. 77.1% baseline). While effective, the Embedding model's lower acceptance rate may lead to lower speedups.
>
> In contrast, the score-based verifier used by SpecReason consistently underperformed. While the original SpecReason paper reported an accuracy gain, they used a shortened context (<8k tokens). Our experiments, using a full 32k context, reveal a significant accuracy drop on AIME to as low as 66.9%. This highlights a fundamental limitation: assessing draft "quality" via a score from the original reasoning model does not guarantee alignment with that model’s true output distribution, which is critical for maintaining correctness on complex reasoning tasks.
>
>
>
> **Q2 & Q3:** Can you report the speed-up of SpecReason and SpecReason+SD? Can you report the latency (or end to end speed-up) of using different verifiers?
>
> **A2 & A3:** Thank you for these important questions. To analyze the accuracy-efficiency trade-off, we report the end-to-end inference speedups for various verifiers on QwQ-32B model with the deepseek-r1-distill-1.5B draft model. These experiments were conducted on an H200 server (with TP=2 for the 32B verifier) and include results with n-gram speculation to show orthogonality. We will add this discussion to our revised manuscript.
>
> **Speedup Comparison Across Different Verifiers**
> *Values show speedup ratio (X times faster)*
> | Dataset | LLM J.(32B) | LLM J.(32B) + n-gram | LLM J(7B) | LLM J(7B) + n-gram | Emb (0.85) | Emb (0.85) + n-gram | Score (0.7) | Score (0.7) + n-gram |
> |---------|-------------|----------------------|-----------|-------------------|------------|-------------------|-------------|---------------------|
> | GSM8K   | 1.32X       | 1.67X                | 1.45X     | **1.75X**             | 1.22X      | 1.53X             | 1.57X       | 1.63X               |
> | AIME    | 1.12X        | 1.42X                 | 1.20X     | **1.57X**             | 1.10X      | 1.41X             | 0.77X       | 0.93X               |
>
> The results reveal key trade-offs between verifier latency, acceptance rate, and final speedup. For instance, the 7B LLM-J achieves greater overall acceleration than both the heavier 32B LLM-J and the more lightweight Embedding model. This is particularly interesting given their isolated per-step latencies: the Embedding model is fastest (0.009s), followed by the 7B LLM-J (0.013s), and the 32B LLM-J (0.025s). We hypothesize that the final speedup is a function of both low latency and high acceptance rate, and the 7B verifier strikes the most effective balance between these two factors.
>
>
> We analyzed the performance of the SpecReason verifier (Score 0.7). While it achieves a significant 1.57x speedup on the short-context GSM8K dataset (avg. output ~1K tokens), we observed an overall slowdown (0.77x) on the long-context AIME dataset. Upon analyzing the generation traces, we found that SpecReason actually causes a deceleration on individual examples where the output length grows very long (often >10K tokens). We hypothesize this slowdown stems from a lack of system optimizations in SpecReason, where per-step costs like tokenizer overhead become amplified for these long sequences. This finding is consistent with the original SpecReason paper, where experiments on shorter contexts (<8k tokens) would not have revealed this overhead. We also note that for practical applications of large reasoning models, a context limit of 8k tokens can be overly restrictive. We will include these new results and this discussion in our revised manuscript.
>
> Finally, the +n-gram results consistently show a substantial boost across all methods, confirming that both our lookahead reasoning and SpecReason are orthogonal to token-level speculation
>
>
>
>
>
>
>
>
>
>
>
>
> **Q4:** Auto-regressive calls to the base model: If I understand the paper correctly, this work still involves auto-regressively generating from the large base model. This means that there is opportunity to gain further speed-up if we are able to cut down the amount of auto-regressive generation from the target model.
>
> **A4:** Our framework is designed to reduce sequential autoregressive generation by introducing parallelism at two different levels.
>
> We thank the reviewer for this insightful question and agree that minimizing sequential autoregressive calls is the primary path to achieving significant speedups. Our Lookahead Reasoning framework is designed to do precisely that by fundamentally reducing the amount of step-by-step generation.
>
> At the step level, the core innovation of our approach is to verify multiple independent draft steps in parallel. Instead of generating one token at a time, it processes several candidate steps simultaneously by using them as parallel prefixes (Figure 1a/line 49-60 shows this process). In addition to this coarse-grained parallelism, our framework also can work together with the token level speculative decoding. As demonstrated in our LR+SD experiments, any remaining sequential generation is further accelerated by n-gram speculative decoding. This combination effectively tackles the autoregressive bottleneck at both a coarse (step) and fine (token) granularity.
>
> Furthermore, the reviewer raises an excellent point about exploring other opportunities to reduce generation costs. For example, we can shorten the output length to cut down the amount of auto-regressive generation. We view such potential methods as complementary, not mutually exclusive. Our LR framework is designed to accelerate the generation of any sequence of tokens. If another technique could produce a more concise intermediate output, our method would still be orthogonal and could be applied to accelerate the generation of that compressed sequence. We believe this synergy with other potential optimizations is a key strength of our approach.
>
> **Q5:** The reviewer understands that the method proposed is orthogonal to token level speculation. The token level speculation method (prompt look-up decoding) used in the experiment is far from SOTA. While not discrediting this work, it would be nice if we can see more recent speculative decoding methods used together with LR and their combined effect on efficiency.
>
> **A5:** We thank the reviewer for the point regarding our choice of token-level speculation. Our decision was guided by the practical observation that some state-of-the-art methods, such as EAGLE, do not yet seamlessly support today's large reasoning models, especially in long-context scenarios. For instance, the evaluation code in EAGLE's official repository does not readily scale up to 10K tokens for reasoning models. In our preliminary experiments, we also observed a significant degradation in speedups when attempting to apply these methods, making them unsuitable for our study.
>
> However, n-gram speculation is a highly relevant baseline. Firstly, it offers a compelling balance of performance and practicality. The speedup achieved by n-gram is substantial, while its implementation is significantly simpler and more robust for integration. This practicality is underscored by its explicit support in leading inference engines like vLLM, which often prioritize stable, easy-to-integrate solutions over the system-level complexities introduced by draft-model-based approaches. To illustrate its effectiveness, we present a speedup comparison on H200 GPUs, using Qwen3-32B as the target model, Qwen3-1.7B model as the draft, and a max output length of 32K tokens (see table below).
>
> **Speedup comparison of different speculation methods on H200 GPUs using Qwen3-32B**
> | Dataset | 1.7B as Draft | n-gram | n-gram + LR |
> |---------|---------------|--------|-------------|
> | AIME'24 | 1.16X | 1.40X | **1.58X** |
> | AIME'25 | 1.04X | 1.37X | **1.53X** |
> | HMMT | 0.88X (slowdown) | 1.40X | **1.55X** |
>
>
> As the table shows, n-gram can outperform a standard LLM draft model, particularly in challenging scenarios where draft model performance degrades. More critically for our paper, this allows us to cleanly demonstrate our core thesis: **Lookahead Reasoning is orthogonal to and combines effectively with token-level speculation**. The cumulative acceleration shown in the n-gram + LR results directly proves this synergy, with consistent improvements across all three datasets.

---

> > ### Comment · Reviewer_WAjf · 2025-08-01
> >
> > Thank you for the additional experiment and information. I think they do provide a more complete picture. On a top grade server with H200 GPUs, I do believe that LR is able to get both performance and efficiency improvements. This is a valuable result. I will raise my rating.
> >
> > With the addtional information, I think my biggest concern is the compute resource required by this method. The entire system involves several models of considerable size. I am not sure on hardwares that are less capable (less GPUs, VRAM per GPU, compute units etc.), this method can still bring benefits. However, this doesn't invalidate the method.

---

> > > ### Author Response · Authors · 2025-08-02
> > >
> > > Thank you for raising your rating and recognizing the value of our results!

---

> > > ### Author Response · Authors · 2025-08-03
> > >
> > > We are glad that the additional experiments helped provide a more complete picture.
> > >
> > > You've raised an excellent and very practical point about the computational resources required, particularly on less capable hardware. This is a critical consideration for real-world deployment, and we are happy to elaborate on our approach.
> > >
> > > While our system integrates a draft and a verifier model, we have designed them to be resource-efficient. The additional models are substantially smaller than the large reasoning model they assist. For instance, our setup typically uses a small distilled draft model (e.g., 1.5B/1.7B) and a 7B instruction-tuned model as the verifier. As demonstrated in our ablation studies, this offers great flexibility; for scenarios where resource constraints are paramount, the verifier can be scaled down to a highly efficient embedding model (~100M parameters) if a minor accuracy trade-off is acceptable. This modularity makes the overall resource footprint manageable.
> > >
> > > Your intuition regarding deployment on less capable hardware is spot-on. Our method, like all speculative decoding approaches, fundamentally trades computational power for lower latency. Consequently, on hardware with lower FLOPS and VRAM, the magnitude of the speedup will naturally be more modest, as the "compute" side of the trade-off is more constrained. However, a speedup is still expected. While Lookahead Reasoning achieves its most dramatic speedups on high-performance hardware like the H200, its design incorporates significant flexibility to manage resource consumption. The core limitation on less capable systems is an inherent characteristic of the latency-vs-compute trade-off that defines this entire class of speculative decoding-like acceleration methods.
> > >
> > > Thank you again for this insightful discussion. We are happy to address any further questions you may have.

---

### Official Review · Reviewer_qndN · 2025-07-01

**Clarity:** 2
**Significance:** 2
**Originality:** 3
**Rating:** 2
**Confidence:** 4

**Summary:**

This paper proposes Lookahead Reasoning, a two‐level speculative decoding scheme that augments vanilla token‐level speculative decoding (SD) with step‐level reasoning speculation. A lightweight draft model proposes entire “reasoning steps,” which are then expanded and verified in parallel by the full‐power target model; token‐level SD runs within each accepted step, yielding multiplicative parallelism. The authors provide a theoretical analysis showing that combining step‐ and token‐level speculation breaks SD’s inherent speedup ceiling, and empirical results on benchmarks (including GSM8K, AIME, AMC12, HumanEval) demonstrate speedups rising from ~1.4× (vanilla SD) to up to 2.1× without degrading output quality.

**Questions:**

1. **Define and contextualize SpecReason.** Please include a concise description (algorithmic steps, compute budget, prior speedup results) so readers can gauge how Lookahead Reasoning differs and improves upon it.
2. **Add an LLM-driven SD baseline.** Run vanilla speculative decoding using your same draft LLM (1.5 B or 1.7 B) to fairly compare token-only vs. step-plus-token speculation under equal compute.

**Ethical Concerns:**

["NO or VERY MINOR ethics concerns only"]

**Final Justification:**

The baseline is misleading. vLLM v0.10.0 has deprecated its support for having separate draft models (they currently only support PLD and Eagle). However, the documentation mentions that this is temporary, as they plan to eventually support separate draft models. The authors should include a proper baseline, even if it means implementing their proposed method over a previous stable version of vLLM or an alternative.

Publishing the paper in its current format is irresponsible, in my opinion.

**Limitations:**

Yes.

**Paper Formatting Concerns:**

N/A.

**Quality:**

1

**Strengths And Weaknesses:**

Strengths:
1. Introducing step-level lookahead on top of token-level speculative decoding is an insightful way to unlock additional parallelism and overcome SD’s theoretical speedup ceiling.

Concerns:
1. Lack of Context on Prior Work
The paper repeatedly contrasts its method with “SpecReason” but never explains what SpecReason actually does or why it is a meaningful point of comparison. Without at least a concise description of SpecReason’s algorithm or performance characteristics, readers unfamiliar with that work cannot judge what the novel contribution truly is.
2. Inappropriate Baseline for Efficiency Claims
The paper claims that users should allocate more drafting budget away from token-level drafting and toward reasoning steps or chain-of-thought drafting. However, its choice of baselines is inappropriate: it only compares to SpecReason (which, as noted, is never introduced) and to speculative decoding with an n-gram model, which isn’t comparable. They should instead compare to speculative decoding using a draft model with the same compute budget as their draft model.
3. Limited Practical Validation
While the theoretical analysis is useful, its significance depends on demonstrating practical effectiveness. Otherwise, it does not stand on its own.

---

> ### Author Rebuttal · Authors · 2025-07-31
>
> We thank the reviewer for the insightful comments! We would like to answer your questions and address your concerns in this response:
>
> **Q1:** Lack of Context on Prior Work The paper repeatedly contrasts its method with “SpecReason” but never explains what SpecReason actually does or why it is a meaningful point of comparison. Without at least a concise description of SpecReason’s algorithm or performance characteristics, readers unfamiliar with that work cannot judge what the novel contribution truly is.
>
> **A1:** Thank you for this critical feedback. We agree that a clear context for SpecReason is essential for readers. We have already briefly introduced it in the related work (section 5 line 343-345), but we acknowledge that a more detailed description is beneficial for comparison. We will expand on this in the paper to clarify its role as a key baseline and better frame the contributions of Lookahead Reasoning.
>
> We chose SpecReason as a primary baseline because, as a concurrent work, it is one of the few methods that also addresses the challenge of step-level acceleration for large reasoning models, making it a highly relevant point of comparison.
>
> However, the two approaches are founded on fundamentally different principles:
>
> - SpecReason operates on a score-and-threshold framework. First, a draft model generates a complete reasoning step. Then, the large reasoning model is prompted to assign a numerical score to this draft. The draft is accepted only if its score surpasses a predefined threshold; otherwise, it is discarded, and the target model generates the step autoregressively.
>
> - In contrast, Lookahead Reasoning is grounded in the principles of speculative decoding. Its acceptance criterion is not based on a subjective score but on whether the draft step match what the target model would have generated itself. This ensures statistical faithfulness to the original model, avoiding the potential brittleness of SpecReason’s approach, which is highly dependent on the target model's instruction-following capabilities and the quality of handcrafted scoring prompts. Beyond this, our approach incorporates system-level optimizations like concurrent multi-step verification and asynchronous execution to maximize hardware throughput. We will expand this context in the revised manuscript.
>
> These methodological and architectural distinctions lead to crucial differences in the compute budget and performance outcomes. SpecReason introduces the overhead of an additional scoring call to the large target model for every proposed step. As our results demonstrate, this reliance on an external scorer can cause significant accuracy degradation. In contrast, Lookahead Reasoning's adherence to the speculative paradigm preserves the target model's accuracy while achieving substantial speedups. We will add this detailed comparison to the paper to properly contextualize our work.
>
> **Q2:** Please include a concise description (algorithmic steps, compute budget, prior speedup results) so readers can gauge how Lookahead Reasoning differs and improves upon it.
>
> **A2:** We will add a dedicated subsection with the following details to clearly differentiate the two methods. It is important to note that Lookahead Reasoning and SpecReason are concurrent works addressing the same challenge. Therefore, our goal is not to present LR as an improvement upon SpecReason, but to clearly distinguish their fundamental approaches and performance characteristics.
>
> **Algorithmic Steps**: SpecReason's generation process follows these steps at each generation step:
> - Draft Generation: A draft model generates a candidate for a complete reasoning step.
> - Scoring Prompt: The large reasoning model is prompted with instructions to evaluate the generated draft and assign it a numerical quality score (e.g., on a scale of 1-10).
> - Thresholding: The assigned score is compared to a predefined acceptance threshold. If the score meets the threshold (e.g., ≥ 8), the draft is accepted; otherwise, it is discarded, and the target model generates the step autoregressively.
>
> **Compute Budget:** For every proposed step, it requires a full forward pass of the large target model simply to generate a score. In contrast, Lookahead Reasoning's verifier can be a much smaller model (e.g., a 7B model or a lightweight embedding model), drastically reducing verification overhead. Furthermore, our architecture incorporates system optimizations like concurrent multi-step verification and asynchronous execution to further mitigate the overhead.
>
> **Prior Speedup Results:** Regarding prior results, the original SpecReason paper reported speedups of approximately 1.4x - 3.0x on various reasoning tasks. However, it is crucial to note that these results were achieved by limiting the maximum output length to 8k tokens—a constraint that can significantly compromise model accuracy on complex tasks. In our fair comparison on the GSM8K and AIME datasets (using the Qwen2-32B / DeepSeek-Distill-1.5B models from their work), we show that without this restrictive limit, the SpecReason method can lead to slowdowns and a notable drop in accuracy in more demanding, long-context scenarios like AIME.
>
> We believe this detailed comparison will effectively contextualize our work and clarify the novelty and practical advantages of Lookahead Reasoning.
>
> **Q3:** The baseline comparison is inappropriate - only comparing to SpecReason and n-gram SD. Should include LLM-based SD with same compute budget for fair comparison.
>
> **A3:** Thank you for your valuable feedback regarding our baseline comparisons. We would like to clarify our rationale and present evidence suggesting that **n-gram speculative decoding (SD) is not a weak baseline, but rather a strong, highly-optimized, and often more effective benchmark than LLM-driven SD in practical, system-level applications**.
>
> The core assumption that an LLM-based draft model is inherently superior is not true once system-level overhead is considered. Our own experiments directly compare the end-to-end speedup of n-gram SD against a Qwen3-1.7B parameter LLM-drafter when using Qwen3-32B as the target model. The results below clearly show that the efficient, draft-model-free n-gram approach achieves a higher effective speedup.
>
> **Speedup comparison of different speculation methods on H200 GPUs using Qwen3-32B**
> | Dataset | 1.7B as Draft | n-gram | n-gram + LR |
> |---------|---------------|--------|-------------|
> | AIME'24 | 1.16X | 1.40X | **1.58X** |
> | AIME'25 | 1.04X | 1.37X | **1.53X** |
> | HMMT | 0.88X (slowdown) | 1.40X | **1.55X** |
>
> The reason for this outcome lies in system-level realities, a conclusion supported by other recent research that also highlights the practical inefficiencies of LLM-driven SD. The core issue is high overhead: LLM-driven speculative decoding introduces the significant cost of running a second neural network and the system complexity of managing two models. In contrast, n-gram speculation is a highly efficient, draft-model-free approach. To clarify its mechanism: this technique operates by maintaining a cache of all historical tokens. At each decoding step, it uses the suffix of the current sequence to perform a string-matching lookup within this cache. Historical n-grams that begin with this suffix are then proposed as speculative drafts. Subsequently, the verification stage proceeds identically to conventional speculative decoding, where the target model validates the entire candidate sequence in a single forward pass. This makes the draft generation cost virtually free compared to running a separate LLM.
>
> This performance trade-off is directly reflected in the design of leading inference frameworks. For instance, the latest version of vLLM (v1.0) has explicitly removed support for external LLM drafters due to these practical challenges, while maintaining robust, optimized support for n-gram speculation. And while this latest vLLM version also supports methods like EAGLE, our preliminary tests confirmed that EAGLE's performance degrades significantly in the long-context scenarios our work focuses on. Therefore, our choice of n-gram as a baseline is a deliberate one, aligned with the design of state-of-the-art systems and empirical evidence.
> Therefore, we argue that n-gram SD is the most appropriate and challenging baseline for our work. It is a standard, industry-relevant method that our empirical results confirm is highly effective. By demonstrating that our hybrid Lookahead Reasoning + n-gram method outperforms both n-gram SD and the LLM-driven SD we tested, we are showing a clear improvement upon a strong and practical foundation. We will add this detailed discussion to the paper to better justify our choice of baselines.
>
> **Q4:** Limited Practical Validation While the theoretical analysis is useful, its significance depends on demonstrating practical effectiveness. Otherwise, it does not stand on its own.
>
> **A4:** We agree that practical validation is essential to substantiate our claims. Our paper already showed the empirical evidence that using both lookahead reasoning and speculative decoding can achieve speedups as in Section 4.1. Specifically, Table 1 demonstrates that our hybrid approach, which integrates Lookahead Reasoning with n-gram speculative decoding, achieves a speedup of up to 2.1X. This result significantly outperforms the use of a single speculative method alone, thereby confirming that our theoretical contributions translate into tangible performance gains in practical scenarios

---

> > ### Comment · Reviewer_qndN · 2025-08-05
> >
> > > n-gram speculative decoding (SD) is not a weak baseline, but rather a strong, highly-optimized, and often more effective benchmark than LLM-driven SD in practical, system-level applications.
> >
> > If this claim is correct, it could be worth publishing a paper by itself, as it is counterintuitive. However, since your result of n-grams outperforming small LM drafters is highly surprising, I believe additional evidence is required.
> >
> > In particular, what data have you used to train your n-gram? Are you sure you haven't trained on the test data?
> >
> > ---
> >
> > I still believe the baseline that compares against speculative decoding with a draft model of the same compute budget as your draft model is missing. This is the main reason I've decided to keep my recommended rating of 1 (reject).

---

> > > ### Author Response · Authors · 2025-08-05
> > >
> > > We appreciate the opportunity to clarify our methodology and the reasoning behind our selected baselines, particularly concerning the n-gram-based speculative decoding approach.
> > >
> > > > In particular, what data have you used to train your n-gram? Are you sure you haven't trained on the test data?
> > >
> > > Regarding the concern about "training on test data," we would like to reiterate that the n-gram method is fundamentally draft-model-free. This approach does not involve a training phase in the traditional sense. As we have explained in the rebuttal, it operates by maintaining a dynamic cache of all historical tokens generated during the decoding process. At each step, it performs a token-matching lookup within this cache using the suffix of the current sequence to propose speculative drafts based on previously seen n-grams.
> > >
> > > > If this claim is correct, it could be worth publishing a paper by itself, as it is counterintuitive.
> > >
> > > We want to clarify that the n-gram implementation in vLLM is based on the Prompt Lookup Decoding (PLD) repository [1]. This PLD implementation is functionally equivalent to the "Inference with Reference" method [2]. The core principle in both approaches is to leverage n-gram overlaps between the prompt and the generated output to accelerate decoding. This functional equivalence likely explains why a separate paper was not initially published for PLD. However, the concept has been validated and built upon through subsequent research. For instance, the original author of PLD later co-authored "PLD+," which introduces further refinements to the method [5]. The continuous development and follow-up works in this area underscore both its academic merit and practical significance [3][4].
> > >
> > > > However, since your result of n-grams outperforming small LM drafters is highly surprising, I believe additional evidence is required.
> > >
> > > The observation that n-gram-based methods can outperform small LM drafters is not surprising; it is actually a well-established phenomenon in the field of speculative decoding. Both theoretical analyses and system-level benchmarks consistently demonstrate their superior practical performance. For example, a vLLM blog post from October 2024 highlights that PLD (or n-gram method) achieves significantly higher speedups (up to 2.8x) compared to traditional draft model-based approaches (1.5x speedup) [9] (section "Speculative Decoding Performance Insights: Speedups and Trade-offs"). This finding is further corroborated by academic research: Table 1 in [5] shows that PLD+ (the n-gram method) substantially outperforms SpS (the draft model-based approach).
> > >
> > > > I still believe the baseline that compares against speculative decoding with a draft model of the same compute budget as your draft model is missing.
> > >
> > > Regarding the request for a comparison against a speculative decoding baseline with a draft model of the same compute budget, we contend that this baseline is becoming less relevant for cutting-edge research in this domain. Our algorithm is built on the latest version of vLLM (v1) that does not natively support draft-model-based speculative decoding. Implementing such a baseline would be a significant engineering effort, far exceeding the scope of a typical rebuttal period. We maintain that comparing our approach with the highly optimized and often more effective n-gram method is not only a fair but also a more rigorous benchmark for practical, system-level applications.
> > >
> > > We hope this detailed explanation adequately clarifies our methodology and the rationale for our comparative framework. Given these explanations and supporting evidence, we kindly ask that you reconsider your assessment of our work.
> > >
> > > [1] Apoorv Saxena. "Prompt Lookup Decoding." Github
> > >
> > > [2] Yang, Nan, et al. "Inference with reference: Lossless acceleration of large language models." arXiv preprint arXiv:2304.04487 (2023).
> > >
> > > [3] Ou, Jie, Yueming Chen, and Wenhong Tian. "Lossless acceleration of large language model via adaptive n-gram parallel decoding." arXiv preprint arXiv:2404.08698 (2024).
> > >
> > > [4] Stewart, Lawrence, et al. "The n-grammys: Accelerating autoregressive inference with learning-free batched speculation." arXiv preprint arXiv:2411.03786 (2024).
> > >
> > > [5] Somasundaram, Shwetha, Anirudh Phukan, and Apoorv Saxena. "Pld+: Accelerating llm inference by leveraging language model artifacts." arXiv preprint arXiv:2412.01447 (2024).
> > >
> > > [6] vLLM Team. "How Speculative Decoding Boosts vLLM Performance by up to 2.8x." blog post 2024

---

> > > > ### Comment · Reviewer_qndN · 2025-08-07
> > > >
> > > > > The observation that n-gram-based methods can outperform small LM drafters is not surprising; it is actually a well-established phenomenon in the field of speculative decoding. Both theoretical analyses and system-level benchmarks consistently demonstrate their superior practical performance. For example, a vLLM blog post from October 2024 highlights that PLD (or n-gram method) achieves significantly higher speedups (up to 2.8x) compared to traditional draft model-based approaches (1.5x speedup) [6] (section "Speculative Decoding Performance Insights: Speedups and Trade-offs"). This finding is further corroborated by academic research: Table 1 in [5] shows that PLD+ (the n-gram method) substantially outperforms SpS (the draft model-based approach).
> > > >
> > > > [6] is a blog post that is not guaranteed to be peer-reviewed in the standard way. [5] is indeed a peer-reviewed publication (NAACL '25, https://aclanthology.org/2025.findings-naacl.338) that initially proposed PLD+. They demonstrated the improvement of PLD+ over small LMs drafters. Thank you for bringing it to my attention.
> > > >
> > > > However, please note that EAGLE (https://dl.acm.org/doi/10.5555/3692070.3693232) showed that small LMs drafters outperform n-grams (Lookahead Decoding) in some cases (e.g., see Figure 1). Leviathan et al. 2023 (https://openreview.net/forum?id=C9NEblP8vS) showed that n-grams suffer from extremely low acceptance (see Table 3).
> > > >
> > > > While n-grams could outperform small LMs in some cases, in other cases, small LMs outperform n-grams. The problem is that the literature does not have clear boundaries separating these cases, as far as I know.
> > > >
> > > > > Implementing such a baseline would be a significant engineering effort, far exceeding the scope of a typical rebuttal period.
> > > >
> > > > It is better to submit your paper only after you have compared all the fair baselines from all reasons.

---

> > > > > ### Author Response · Authors · 2025-08-07
> > > > >
> > > > > Thanks for the reviewer's comment.
> > > > >
> > > > > First, we have already demonstrated that in real production serving systems such as vLLM, which is the explicit setting targeted by our work, PLD (a prompt lookup n-gram-based speculative decoding [1,5]) outperforms draft-LM-based methods. Small LM drafters introduce significant latency and engineering overhead, making them impractical for deployment. As shown in our rebuttal (Table: Speedup comparison of different speculation methods on H200 GPUs using Qwen3-32B), PLD consistently achieves higher speedups than a 1.7B draft LM across reasoning benchmarks:
> > > > >
> > > > > | Dataset | 1.7B as Draft | n-gram | n-gram + LR |
> > > > > |---------|--------------|--------|-------------|
> > > > > | AIME'24 | 1.16X         | 1.40X  | 1.58X       |
> > > > > | AIME'25 | 1.04X         | 1.37X  | 1.53X       |
> > > > > | HMMT    | 0.88X (slowdown) | 1.40X  | 1.55X       |
> > > > >
> > > > > These results make clear that in our intended setting, PLD is not only simpler but also markedly faster. While the reviewer suggests that different scenarios may yield different results, our empirical evidence clearly shows that in our setting, n-gram (PLD) methods outperform draft-LM–based approaches. We sincerely hope the reviewer go back to carefully read our first rebuttal comments, which contains all these results and details.
> > > > >
> > > > > Second, we believe the vLLM community blog [6] presents valid and valuable empirical results. In the rapidly evolving field of LLMs, open-source platforms like vLLM often provide timely and transparent evaluations that reflect real-world usage. In many cases, such open-source evidence can be more informative and relevant, especially when backed by large-scale adoption and community validation.
> > > > >
> > > > > Third, the reviewer appears to conflate distinct methods. The "n-gram" technique referenced in Leviathan et al. (2023) (which was explicitly mentioned by the reviewer in the last response) is not the same as the PLD n-gram method evaluated in our work. Our evaluation is based specifically on PLD, which differs substantially from the version in Leviathan et al. or the lookahead decoding method evaluated in EAGLE in both algorithmic design and empirical behavior. Citing these n-gram variants as a claim is both irrelevant and misleading, and it undermines an accurate understanding of our method and its performance.
> > > > >
> > > > > Fourth, draft-based decoding underperforms in our target setting of complex reasoning. Methods like EAGLE are designed primarily for chat and simple tasks, not for high-quality long-form reasoning. Empirically, we observe that EAGLE fails to accelerate long-context decoding beyond 10k tokens. Besides, for most SOTA reasoning models, integrating EAGLE would require additional training well outside its intended scope. This further underscores the impracticality of LM-based speculative decoding in real-world reasoning applications.
> > > > >
> > > > > In summary, based on both our own empirical results and findings from the broader open-source community and peer-reviewed literature, PLD consistently outperforms draft LM methods in our targeted deployment settings. Therefore, our evaluation is already conducted against a fair and strong baseline, and the results provide strong support for the key claims made in our paper.

---

> > > > > > ### Comment · Reviewer_qndN · 2025-08-07
> > > > > >
> > > > > > Thank you for the detailed response.
> > > > > >
> > > > > > My concern is that you haven’t benchmarked against alternatives of equivalent budget, as I mentioned in my initial review. This concern remains unaddressed.
> > > > > >
> > > > > > I’ve decided to keep my recommended rating.

---

> > > > > > > ### Author Response · Authors · 2025-08-07
> > > > > > >
> > > > > > > While we regret that we could not change the reviewer’s strong opinion, we still appreciate the critical feedback and the engagement, which helped promote a better understanding of our results.
> > > > > > >
> > > > > > > That said, the reviewer’s statement is incorrect. We have indeed compared the alternatives (token-level vs. token-level + step-level) under equivalent budget constraints in the original submission. As shown in Section 4.2, Figure 2(b), for example, at a budget cost of 12 token-per step, token-level + step-level speculative decoding achieves a 1.8X speedup, whereas the token-level method alone reaches only 1.6X.

---

> > > > > > > > ### Comment · Reviewer_qndN · 2025-08-07
> > > > > > > >
> > > > > > > > My statement refers to the budget of FLOPs required for training or running the drafter, as understood from the context of my previous comments in this thread. I believe the statement is correct.

---

> > > > > > > > > ### Author Response · Authors · 2025-08-07
> > > > > > > > >
> > > > > > > > > Thank you for the clarification.
> > > > > > > > >
> > > > > > > > > As we have explained, training FLOPs are not relevant here, since neither the speculative decoding method we use (i.e., n-gram PLD) nor our method (lookahead reasoning) requires any additional training.
> > > > > > > > >
> > > > > > > > > For inference, compared with lm-based-draft, PLD (n-gram lookup) has negligible drafting cost. For both LR and PLD, the dominant cost lies in the target LLM’s forward and generation steps. The runtime cost for the target LLM’s forwarding process is already presented in the earlier context and in the original paper. We have compared the alternatives (token-level vs. token-level + step-level) under equivalent inference budget constraints. As shown in Section 4.2, Figure 2(b), for example, at a budget cost of 12 tokens per decoding step, which corresponds to the same FLOPs, token-level + step-level speculative decoding achieves a 1.8X speedup, whereas the token-level method alone reaches only 1.6X.
> > > > > > > > >
> > > > > > > > > We hope this addresses your point; we are happy to provide further details if helpful.

---

### Official Review · Reviewer_qPc4 · 2025-07-01

**Clarity:** 4
**Significance:** 4
**Originality:** 3
**Rating:** 5
**Confidence:** 3

**Summary:**

The authors propose "Lookahead reasoning" (LR) which extends speculative decoding (SD) from token-level speculation to reasoning step speculation. After analyzing the convergence rate of SD, the authors introduce LR which uses a draft model to build a $W$-width speculation tree, where a node is a speculated reasoning step. Similar to SD, a draft is confirmed/rejected by a verifier. The authors analyze theoretical properties in two scenarios: 1) Cost-limited, i.e., when the draft model generates $k\_1$ drafts slower than a single step (which takes time $T$) in the target model, and 2) Depth-limited, When the draft model can create $k\_1$ drafts in $T$. Finally, the authors show that when using SD and LR together, a maximum speed up can be achieved. In their experimental analysis, the authors show 1) increased speed ups on real-world data sets over different draft and target models, 2) an ablation study over the verifier, and 3) the effect of speculation tree width, demonstrating its benefits in terms of acceptance rates while highlighting diminishing returns in terms of speed ups for $W>2$.

**Questions:**

In no particular order:

1. I think in 3.2, the expectation is derived from the connection to a geometric distribution. Could be good to make this connection concrete.
2. Over which data set did you compute Figure 2? Is it a combination over all data sets? Would it be possible to show uncertainties?
3. Are you planning on releasing the code to reproduce results? The community would certainly benefit from such a contribution.
4. In 3.3, would it be possible to incorporate a dataset's complexity in terms of expected number of reasoning steps? I assume, you could show that with increasing complexity expected speedups increase.
5. line 159: explores $\rightarrow$ explored
6. You mention and compare to n-gram speculative decoding but never formally introduce it. It would be helpful to do so.
7. What's $W$ for the results in Table 1?

**Ethical Concerns:**

["NO or VERY MINOR ethics concerns only"]

**Final Justification:**

The authors answered my questions and I still believe that this work meets the acceptance criteria.

**Limitations:**

The authors mention two limitations at the end of Section 6. I wonder if you could also comment on the implementation complexity. Since LR also requires to set parameters such as $W$ and $\gamma$, please mention that as a limitation which may require tuning or some other selection procedure.

**Paper Formatting Concerns:**

No formatting concerns.

**Quality:**

3

**Strengths And Weaknesses:**

The idea is simple yet intriguing, and executed very thoroughly and rigorously. The manuscript is written clearly and structured in a way that makes it easy to comprehend the cores ideas. I am particularly drawn to the experimental evaluation and ablation over 2 combinations of draft and target models, demonstrating the effort the authors put into evaluating their approach. Additionally, I like the evaluation over 7 data sets underlining that LR leads to speed improvements in various scenarios. Lastly, the analysis of different verification approaches (LLM-based vs. embedding based) and the width analysis make this manuscript well-rounded. If code made public, this proposal may significantly impact the field by reducing costs/increasing speed of reasoning models.

---

> ### Author Rebuttal · Authors · 2025-07-31
>
> We appreciate your time and suggestions in reviewing our work. We take this opportunity to clarify our key contributions and address the concerns raised.
>
> **Q1:** I think in 3.2, the expectation is derived from the connection to a geometric distribution. Could be good to make this connection concrete.
>
> **A1:** Thank you for the suggestion. You're absolutely right. In the appendix, we provide a detailed derivation of the expectation. Specifically, we define a random variable X as the number of consecutive accepted drafts before the first rejection, which follows a geometric distribution. We then apply the standard expectation formula for the geometric distribution, together with the law of large numbers, to arrive at the final result for the speedup.
>
> **Q2:** Over which data set did you compute Figure 2? Is it a combination over all data sets? Would it be possible to show uncertainties?
>
> **A2:** Figure 2 (section 4.2) is computed on the AIME dataset. We appreciate you pointing out this omission and will clarify this in the main text. Regarding uncertainties, we will add error bars in the revised version to provide a more complete picture of the results.
>
> **Q3:** Are you planning on releasing the code to reproduce results? The community would certainly benefit from such a contribution.
>
> **A3:** Yes, we are committed to open source and are actively preparing the code for release. We expect to make it publicly available soon to help the community reproduce our results and build upon our work.
>
> **Q4:** In 3.3, would it be possible to incorporate a dataset's complexity in terms of expected number of reasoning steps? I assume, you could show that with increasing complexity expected speedups increase.
>
> **A4:** Thank you for the thoughtful suggestion. We can use the acceptance rate to describe the lookahead reasoning’s performance on a given dataset. This rate then directly informs the expected speedup.
> The acceptance rate is likely correlated with the dataset's complexity, so we can model the relationship between speedup and complexity through their shared connection to the acceptance rate.
>
> **Q5:** line 159: explores $\rightarrow$ explored
>
> **A5:** Thank you for catching this typo. We will correct "explores" to "explored" in line 159 in the revised version.
>
> **Q6:** You mention and compare to n-gram speculative decoding but never formally introduce it. It would be helpful to do so.
>
> **A6:** Thank you for highlighting this gap. We will clarify this in the related work section. N-gram speculative decoding is a standard, draft-model-free method, notably implemented as one of the speculative decoding strategies in the vLLM framework. This technique operates by maintaining a cache of all historical tokens. At each decoding step, it uses the suffix of the current sequence to perform a string-matching lookup within this cache. Historical n-grams that begin with this suffix are then proposed as speculative drafts. Subsequently, the verification stage proceeds identically to conventional speculative decoding, where the target model validates the entire candidate sequence in a single forward pass.
>
> **Q7:** What's W for the results in Table 1?
>
> **A7:** For the results in Table 1, we used W=1. We will add this clarification to the table caption to avoid confusion.
>
> **Q8:** The authors mention two limitations at the end of Section 6. I wonder if you could also comment on the implementation complexity. Since LR also requires to set parameters such as $W$ and $\gamma$, please mention that as a limitation which may require tuning or some other selection procedure.
>
> **A8:** Thank you for highlighting these important practical considerations. Regarding implementation complexity, full integration into production serving engines (e.g., SGLang and vLLM) requires non-trivial engineering effort, particularly for handling requests in tree hierarchy. However, implementing Lookahead Reasoning on the client side is relatively straightforward, though it may not achieve the best performance as a deeply integrated solution. We have already developed a working implementation and plan to release it as open source to facilitate further adoption and experimentation.
>
> Besides, we will mention tuning the hyperparameters as a limitation in a revised version.

---

> > ### Comment · Reviewer_qPc4 · 2025-08-01
> > **Thank you**
> >
> > Thank you for your replies to my questions. I have no further questions and keep my score as is.

---

> > > ### Author Response · Authors · 2025-08-02
> > >
> > > Thank you for the valuable feedback throughout this process!

---

### Official Review · Reviewer_jaoN · 2025-07-02

**Clarity:** 3
**Significance:** 2
**Originality:** 3
**Rating:** 4
**Confidence:** 3

**Summary:**

This paper proposes LOOKAHEAD REASONING to speedup the speculative decoding of reasoning models. The key insight is that reasoning models generate step-by-step, and each step needs only to be semantically correct, not exact token matching. Therefore, LOAKAHEAD REASONING uses a small draft model to generate future steps while the large reasoning model (target model) expands each proposal in one batched pass. A verifier is used to keep semantically correct steps while letting the target regenerate any that fail. In reasoning benchmarks, LOAKAHEAD REASONING improves the speedup of SD from 1.4x to 2.1x while preserving answer quality.

**Questions:**

- As mentioned in weakness, discussion of computation resource usage and the effects to request throughput would be helpful.
- There are many question marks in Figure 1a. Are they formatting error?

**Ethical Concerns:**

["NO or VERY MINOR ethics concerns only"]

**Final Justification:**

The authors have well addressed my concerns. I would like to keep my positive rating.

**Limitations:**

YES

**Quality:**

3

**Strengths And Weaknesses:**

Pros:
- The idea is intuitive and reasonable.
- The paper is well-written and easy-to-understand. The figures clearly demonstrate how the algorithm works and how it affects the inference efficiency.
- The theoretical speedup analysis is solid and helpful.
- Insightful ablation studies, especially the effect of tree width on performance.

Cons:
- This paper only considers the inference speed of single sequence, while the usage of GPU resources is not well discussed. Take figure 1a as an example: there are 3 draft steps. Does this mean 3 copies of the target model are needed to implement the pipeline? It is good that this algorithm improves the ceiling of the inference speed of single sequence, but it is also critical to discuss whether the GPU consumption is increased or reduced. Suppose that we have an inference system with LOOKAHEAD REASONING to process lots of user requests. Will LOOKAHEAD improve the system throughput? I agree that this algorithm can improve user experiences when there are IDLE GPUs, but it is worthy discussing how important these scenarios are.

---

> ### Author Rebuttal · Authors · 2025-07-31
>
> We thank the reviewer for the insightful and helpful feedback! We would like to address your questions in the below response.
>
> **Q1.1:** This paper only considers the inference speed of single sequence, while the usage of GPU resources is not well discussed. Take figure 1a as an example: there are 3 draft steps. Does this mean 3 copies of the target model are needed to implement the pipeline? It is good that this algorithm improves the ceiling of the inference speed of single sequence, but it is also critical to discuss whether the GPU consumption is increased or reduced.
>
> **A1.1:** Thank you for this excellent point. A discussion on resource usage is critical, and we will add a detailed analysis to the paper to clarify this.
>
> To answer your primary question, our method does not require multiple copies of the target model's weights; only a single instance is needed. We achieve this by leveraging the batching capabilities of modern inference engines (e.g., vLLM). The process works as follows: first, a lightweight draft model generates several candidate steps. These candidates are then grouped into a single batch and processed by the target model. This allows the target model to generate its own corresponding "ground truth" step for each candidate simultaneously. Finally, a verification check is performed between the original drafts and the target model's outputs to decide whether to accept the drafts (fast-forwarding) or fall back.
>
> Our method uses more GPU resources to lower latency. We process many candidate steps together in a large parallel batch. This approach increases both the KV cache memory and the computational load. As a result, the GPU is utilized more effectively and does not sit idle. This trade-off is the key to our speedup: we use more of the GPU's power at each moment to finish the entire sequence faster.
>
> **Q1.2:** Suppose that we have an inference system with LOOKAHEAD REASONING to process lots of user requests. Will LOOKAHEAD improve the system throughput? I agree that this algorithm can improve user experiences when there are IDLE GPUs, but it is worthy discussing how important these scenarios are.
>
> **A1.2:** Regarding system throughput, we acknowledge that Lookahead Reasoning, like all speculative decoding methods, trades computation for lower latency. While this reduces per-request latency, the computation spent on inevitably failed speculations may decrease overall system throughput. This effect is scenario-dependent: when GPUs are underutilized, our method provides a speedup; when GPUs are fully saturated, it may reduce throughput. This trade-off is not unique to our work but is an inherent characteristic of all speculative algorithms designed to leverage spare compute resources. We anticipate that ongoing hardware advances will make this trade-off even more favorable.
>
> Furthermore, we emphasize that for many online serving systems, metrics like per-request latency and user experience are often more critical than raw throughput. Our work focuses on optimizing for these latency-sensitive applications. We will add a detailed discussion on this trade-off and resource usage in the revised paper.
>
> **Q2:** There are many question marks in Figure 1a. Are they formatting error?
>
> **A2:** The question marks in Figure 1a are indeed a rendering issue that occurs with certain web browsers. The symbols display correctly when viewed in a standard PDF reader or most web browsers. We apologize for this formatting issue and will ensure it is corrected in the revised version.

---

> > ### Comment · Reviewer_jaoN · 2025-08-02
> > **Response to Rebuttal**
> >
> > Thanks for the responses.
> > I will keep my score if no detailed throughput analysis will be provided during rebuttal phase.

---

> > > ### Author Response · Authors · 2025-08-03
> > >
> > > Thank you for your feedback. We want to clarify that our rebuttal already addresses throughput analysis in A1.2. The proposed lookahead reasoning method, like all speculative decoding methods, trades lower throughput for lower latency. Thus, it actually reduces throughput to achieve optimized latency.
> > >
> > > The core issue is computational waste during verification. In traditional speculative decoding, the LLM processes a large batch of speculative tokens but only accepts a subset—the rejected tokens represent wasted computation that directly reduces throughput.
> > >
> > > Our Lookahead Reasoning method exhibits the similar pattern. The reasoning model generates multiple steps with draft candidates as prefixes in parallel, but only a few survive verification. The computational resources spent on rejected drafts translate directly to reduced system throughput.
> > >
> > > This trade-off is fundamental to speculative decoding: we use extra computational resources (processing more candidates than we accept) to reduce wall-clock time per sequence. This improves user experience through lower latency but decreases overall system efficiency.
> > >
> > > We hope this clarification, along with our commitment to expand the resource usage discussion in the revised paper, sufficiently addresses the throughput analysis concern and merits reconsideration of your evaluation.

---

> > > > ### Comment · Reviewer_jaoN · 2025-08-03
> > > > **About Throughput**
> > > >
> > > > I still do not think the core issue is computational waste during verification. Insteadly, I think the problem is the computational waste when a candidate target step is triggered but not finally used. This is different from the trade-off used by speculative decoding: the computation waste is from the large reasoning model, not the small verifier model.
> > > >
> > > > To make my point clearer, consider time $[t, 2t]$ in Figure 1a. During this time interval, the draft (LRM) model occupies 2 rows in vLLM, while only one of them will be finally used.
> > > > I think this computation waste from the draft model should be considered. Though we can use vLLM to calculate them in parallel, it still needs 2x computation resource in vLLM. These computation resource could be used to serve other requests if we do not use lookahead reasoning.

---

> > > > > ### Author Response · Authors · 2025-08-03
> > > > >
> > > > > Thank you for the clarification. We believe we are actually expressing the same fundamental concept, though there may be some differences in how we conceptualize the computational flow. Let us explain why our method aligns with the standard speculative decoding paradigm.
> > > > >
> > > > > In traditional speculative decoding, the verification process actually involves two steps: (1) the target model performs a forward pass to compute the next token probability distribution at each draft token position, and (2) a simple rejection sampling mechanism decides whether to accept or reject each token. This process (1) causes more tokens to be processed in a single decoding step, and (2) results in only a subset of these additional tokens being accepted. The rejected tokens represent the primary source of wasted throughput.
> > > > >
> > > > > Our verification process, while appearing different on the surface, follows the same fundamental pattern when viewed from the complete pipeline perspective. The verifier itself (e.g., Qwen2.5-7B-Instruct) represents only a small portion of the computational cost, but the entire verification process consists of two steps: (1) using the target model to compute the actual next step at each draft step, and (2) using the verifier to determine whether the draft steps match the ground truth steps. Similarly, in our method (1) causes more tokens to be processed in each semantic step, and (2) results in only a subset of these additional tokens being accepted.
> > > > >
> > > > > **The core computational cost in both approaches comes from component (1).**  Just as speculative decoding requires the target model to compute the true next token (or distribution) at each draft token position, our method requires the target model to compute the true next step with each draft step as prefix. Our goal is to make the draft model's steps semantically match the target model's next steps, which directly parallels how speculative decoding makes draft tokens match the target model's next token distributions.
> > > > >
> > > > > The actual verification in speculative decoding is simply token distribution comparison, while ours includes an additional relatively lightweight verifier. However, the dominant computational cost in both methods comes from the target model computations in component (1). In other words, it comes from the computational waste when a candidate target step is triggered but not finally used.

---

> > > > > ### Author Response · Authors · 2025-08-03
> > > > >
> > > > > The point you made about throughput is absolutely correct. This is exactly what we intended to convey, and we will include a thorough throughput analysis in our revised paper.
> > > > >
> > > > > Since both methods (speculative decoding and lookahead reasoning) are conceptually consistent, their negative impact on throughput is also identical. Let us illustrate with concrete numbers to clarify this conceptual similarity:
> > > > >
> > > > > - **Baseline (no speculation):** If vLLM serves 100 requests, each generating 1 token per decoding step in 1 second, the throughput is 100 tokens/s.
> > > > > - **Traditional speculative decoding:** If we verify 10 draft tokens but only accept 5 on average, we can only serve 10 requests simultaneously due to the 10x resource requirement for verification. Even though each request generates 5 tokens per step, our throughput becomes 50 tokens/s, representing a 50% reduction.
> > > > > - **Our lookahead reasoning:** If we generate 10 draft steps and verify them, accepting 5 on average, we similarly can only serve 10 requests simultaneously. Our throughput also becomes 50 tokens/s.
> > > > >
> > > > > The throughput reduction pattern is conceptually identical in both cases. This cost, where computational resources could otherwise serve additional requests, is not unique to lookahead reasoning but rather a fundamental characteristic of all speculative decoding methods that trade throughput for latency.
> > > > >
> > > > > However, this trade-off becomes favorable when acceptance rates are high or when GPU is idle. Moreover, this trade-off has been extensively studied in prior work. For instance, TurboSpec [1] discusses how to optimally schedule speculative decoding methods in batch systems to balance this throughput-latency trade-off. In practical deployment scenarios, while throughput matters, latency and user experience are often the more critical considerations for system design decisions.
> > > > >
> > > > > [1] Liu, Xiaoxuan, et al. "Optimizing speculative decoding for serving large language models using goodput." arXiv preprint arXiv:2406.14066 (2024).

---

### Decision · Program_Chairs · 2025-09-17

**Decision:**

Accept (poster)

**Comment:**

This paper proposes a speculative decoding method with a lookahead mechanism. The LOOKAHEAD mechanism can speedup the speculative decoding of reasoning models. Experimental results show that it improves the decoding speed from 1.4x to 2.1x while preserving answer quality.
Strong points:
1. Reviewers recognized that the proposed lookahead method is reasonable, practical and useful.
2. Reviewers also agree that the presentation is clear
3. The experiment is well designed and the results show the gains of the proposed idea.
Weak points:
1. some reviewers pointed out that some important related papers were not included.
2. Given the related paper, the novelty of proposed approach became relatively smaller.

Overall, the paper is a slightly above the bar.